# Demystifying the Optimization and Generalization of Deep PAC-Bayesian Learning

## Abstract

PAC-Bayes has long been a generalization analysis framework where the expected population error can be bounded by the sum of training error and the divergence between posterior and prior distribution. In addition to being a successful generalization bound analysis tool, the PAC-Bayesian bound can also be incorporated into an objective function to train a probabilistic neural network, which we refer to simply as *PAC-Bayesian Learning*. PAC-Bayesian learning has been proven to be able to achieve a competitive expected test set error numerically, while providing a tight generalization bound in practice, through gradient descent training. Despite its empirical success, the theoretical analysis of deep PAC-Bayesian learning for neural networks is rarely explored. To this end, this paper proposes a theoretical convergence and generalization analysis for PAC-Bayesian learning. For a deep and wide probabilistic neural network, we show that when PAC-Bayesian learning is applied, the convergence result corresponds to solving a kernel ridge regression when the probabilistic neural tangent kernel (PNTK) is used as its kernel. Based on this finding, we further obtain an analytic and guaranteed PAC-Bayesian generalization bound for the first time, which is an improvement over the Rademacher complexity-based bound for deterministic neural networks. Finally, drawing insight from our theoretical results, we propose a proxy measure for efficient hyperparameter selection, which is proven to be time-saving on various benchmarks.

## 1 Introduction

Deep learning has demonstrated powerful learning capability due to its over-parameterization structure, in which various network architectures have been responsible for its significant leap in performance (LeCun et al., 2015). Over-fitting and complex hyperparameters are two of the major challenges in deep learning, hence designing generalization guarantees for deep networks is an important research goal (Zhang et al., 2021). Recently, a learning framework that trains a probabilistic neural network with a PAC-Bayesian bound objective function has been proposed (Bégin et al., 2016; Dziugaite & Roy, 2017; Neyshabur et al., 2017b; Raginsky et al., 2017; Neyshabur et al., 2017a; London, 2017; Smith & Le, 2017; Pérez-Ortiz et al., 2020; Guan & Lu, 2022), which is known as *PAC-Bayesian learning*. While providing a tight generalization bound, PAC-Bayesian learning has been proven to be able to achieve a competitive expected test set error (Ding et al., 2022). Furthermore, this generalization bound computed from the training data can obviate the need for splitting data into training, testing, and validation set, which is highly applicable for training a deep network with scarce data (Pérez-Ortiz et al., 2020; Grünwald & Mehta, 2020). Meanwhile, these advancements on PAC-Bayesian bounds have been widely adapted with different deep neural network structures including convolutional neural network (Zhou et al., 2018; Pérez-Ortiz et al., 2020), binary activated multilayer networks (Letarte et al., 2019), partially aggregated neural networks (Biggs & Guedj, 2020), and graph neural network (Liao et al., 2020).

Due to the impressive empirical success of PAC-Bayesian learning, there is increasing interest in understanding its theoretical properties. However, it is either restricted to a specific technique variant such as Entropy-SGD which minimizes an objective indirectly by approximating stochastic gradient ascent on the so-called local entropy (Dziugaite & Roy, 2018a) and differential privacy (Dziugaite & Roy, 2018b), or relies heavily on empirical exploration (Neyshabur et al., 2017a; Dziugaite et al., 2020). To our best knowledge, there has been no investigation so far into why the training of

PAC-Bayesian learning is successful and why the PAC-Bayesian bound is tight on unseen data after training. For example, it is still unclear when applying gradient descent to PAC-Bayesian learning:

**Q1:** How effective is gradient descent training on a training set?

**Q2:** How tight is the generalization bound compared to those learning frameworks using non-probabilistic neural networks?

The answers to these questions can be highly non-trivial due to the inherent non-convex problem of over-parameterization (Jain & Kar, 2017) and additional randomness introduced by probabilistic neural networks (Specht, 1990) as well as additional challenges brought by the divergence between posterior/prior distribution pairs known as Kullback-Leibler (KL) divergence. Nevertheless, this paper shows that it is possible to answer the above questions by leveraging the recent advances in deep learning theory with over-parameterized setting. It has been shown that wide networks optimized with gradient descent can achieve a near-zero training error, and the critical factor that governs the training process is the neural tangent kernel (NTK), which can be proven to be unchanged during gradient descent training (Jacot et al., 2018), thus providing a guarantee for achieving a global minimum (Du et al., 2019; Allen-Zhu et al., 2019). Under the PAC-Bayesian framework, NTK is no longer calculated from the derivative of the weights directly, but instead is calculated based on the gradient of the distribution parameters of the weights. We call this Probabilistic NTK (PNTK), based on which we build a convergence analysis to characterize the optimization process of PAC-Bayes learning. Due to the explicit solution obtained by optimization analysis, we further formulate the generalization bound of PAC-Bayesian learning for the first time, and demonstrate its advantage by comparing it with the theoretical generalization bound of learning framework with non-stochastic neural networks (Arora et al., 2019a; Cao & Gu, 2019; Hu et al., 2019).

We summarize our contributions as follows:

- With a detailed characterization of gradient descent training of the PAC-Bayes objective function, we derive that the final solution is kernel ridge regression with its kernel being the PNTK.

- Based on the optimization solution, we derive an analytical and guaranteed PAC-Bayesian bound for deep networks for the first time. Moreover, our bound differs from other PAC-Bayes bounds. Recent papers require distribution of posterior, while our bound is completely independent of computing the distribution of posterior.

- The performance of PAC-Bayesian learning depends on the selection of a large number of hyperparameters. We design a training-free proxy based on our theoretical bound and show it is effective and time-saving.

- Our technique of analyzing optimization and generalization of probabilistic neural networks through over-parameterization has a wide range of applications such as the Variational Auto-encoder (Kingma & Welling, 2013; Rezende et al., 2014) and deep Bayesian networks (MacKay, 1992; Neal, 2012), we believe our technique can provide the basis for the analysis of over-parameterized probabilistic neural networks.

## 2 RELATED WORK

**PAC-Bayesian analysis.** A Probably Approximately Correct (PAC) Bayes framework (McAllester, 1999a;b) can incorporate knowledge about the learning algorithm and probability distribution over a set of hypotheses, thus providing a test performance (generalization) guarantee. Subsequently, the PAC-Bayesian method is adopted to analyze the generalization bound of the probabilistic neural networks (Langford & Caruana, 2002b). While the original PAC-Bayes theory only works with a bounded loss function, Haddouche et al. (2021) expanded the PAC-Bayesian theory to learning problems with unbounded loss functions. Furthermore, several improved PAC-Bayesian bounds suitable for different scenarios are introduced by Bégin et al. (2014; 2016). As a result of the flexibility and generalization properties of PAC-Bayes, it is widely used to analyze complex, non-convex, and overparameterized optimization problems, especially over-parameterized neural networks (Guedj, 2019). Neyshabur et al. (2017b) presented a generalization bound for feedforward neural networks with ReLU activations in terms of the product of the spectral norm of the layers and the Frobenius norm of the weights.

**PAC-Bayesian learning.** In addition to obtaining the theoretical analysis for the generalization properties of deep learning, it is important to achieve a numerical bound on generalization for practical deep learning algorithms. Langford & Caruana (2002a) introduced a method to train a Bayesian neural network and used a refined PAC-Bayesian bound for computing the error upper bound. Later, Neyshabur et al. (2017a) extended Langford et al. Langford & Caruana (2002a)'s work by developing a training objective function derived from a relaxed PAC-Bayesian bound. In the standard application of PAC-Bayes, the prior is typically chosen to be a spherical Gaussian centered at the origin. However, without incorporating the information of data, the KL divergence might be unreasonably large, limiting the performance of the PAC-Bayes method. To address this gap, a large volume of literature proposes to obtain localized PAC-Bayes bounds via distribution-dependent priors through data (Ambroladze et al., 2007; Negrea et al., 2019; Dziugaite et al., 2020; Perez-Ortiz et al., 2021). Furthermore, Dziugaite & Roy (2018b); Tinsi & Dalalyan (2022) showed how a differentially private data-dependent prior yields a valid PAC-Bayes bound for a situation where the data distribution is presumed to be unknown. More recently, research has focused on providing a PAC-Bayesian bound for more realistic architectures, such as convolutional neural network (Zhou et al., 2018) , binary activated multilayer networks (Letarte et al., 2019), partially aggregated neural networks (Biggs & Guedj, 2020), and graph neural networks (Liao et al., 2020). We denote the practical use of the PAC-Bayesian algorithm to train over-parameterized neural networks as PAC-Bayesian learning and the target of this work is to demystify the success behind deep learning trained via the PAC-Bayesian bound through PNTK.

## 3 PRELIMINARY

**Notation.** We use bold-faced letters for vectors and matrices and non-bold-faced letters for scalars. We use $\|\cdot\|_2$ to denote the Euclidean norm of a vector or the spectral norm of a matrix, while denoting $\|\cdot\|_F$ as the Frobenius norm of a matrix. For a neural network, we denote $\sigma(x)$ as the activation function. We denote $[n] = \{1, 2, \ldots, n\}$. The least eigenvalue of matrix $\mathbf{A}$ is denoted as $\lambda_0(\mathbf{A}) = \lambda_{\min}(\mathbf{A})$.

### 3.1 DEEP PROBABILISTIC NEURAL NETWORK

In PAC-Bayesian learning we use probabilistic neural networks (PNNs) instead of deterministic networks, where the weights always follow a certain distribution. In this work, we adopt the Gaussian distribution for the weights, and define a $L$-layer probabilistic neural network governed by the following recursive expression,

$$\mathbf{x}^{(l)} = \frac{1}{\sqrt{m}}\sigma\big(\mathbf{W}^{(l)}\mathbf{x}^{(l-1)}\big), 1 \le l \le L; \quad f = \mathbf{v}^\top\mathbf{x}^{(L)} \tag{1}$$

where $\mathbf{x}^{(0)} = \mathbf{x} \in \mathbb{R}^d$ is the input, $\mathbf{W}^{(1)} \in \mathbb{R}^{m \times d}$ is the weight matrix at the first layer, $\mathbf{W}^{(l)} \in \mathbb{R}^{m \times m}$ is the weight at the $l$-th layer for $2 \le l \le L$, and $\mathbf{v} \in \mathbb{R}^m$ is the weight vector at the output layer. To keep weights follow Gaussian distribution during gradient descent training, we introduce the re-parameterization trick (Kingma & Welling, 2013; Kingma et al., 2015):

$$\mathbf{W}^{(l)} = \mathbf{W}_\mu^{(l)} + \mathbf{W}_\sigma^{(l)} \odot \boldsymbol{\xi}^{(l)}, \ \boldsymbol{\xi}^{(l)} \sim \mathcal{N}(\mathbf{0}, \mathbf{I}), \ 1 \le l \le L; \quad \mathbf{v} = \mathbf{v}_\mu + \mathbf{v}_\sigma \odot \boldsymbol{\xi}^{(v)}, \ \boldsymbol{\xi}^{(v)} \sim \mathcal{N}(\mathbf{0}, \mathbf{I}), \tag{2}$$

where $\odot$ denotes the element-wide product operation, thus $\boldsymbol{\xi}^{(l)}$ for $1 \le l \le L$ and $\boldsymbol{\xi}^{(v)}$ share the same size as their corresponding weight matrix or vector. The key insight of re-parameterization is to sample $\boldsymbol{\xi}^{(l)}$ for $1 \le l \le L$ and $\boldsymbol{\xi}^{(v)}$ from a normal distribution $\mathcal{N}(\mathbf{0}, \mathbf{I})$, and leave $\mathbf{W}_\mu^{(l)}$, $\mathbf{W}_\sigma^{(l)}$, $\mathbf{v}_\mu$, and $\mathbf{v}_\sigma$ to be deterministic.

We adopt random initialization for mean weights, where $\mathbf{W}_\mu^{(l)}, \mathbf{v}_\mu \sim \mathcal{N}(\mathbf{0}, c_\mu^2 \cdot \mathbf{I})$ for $l \in [1, L]$. With an abuse of notation, we omit the size of mean $\mathbf{0}$ and variance $\mathbf{I}$ which is in coordinate with their weight matrix or vector. On the other hand, we use an absolute constant to initialize variance weights, namely $\mathbf{W}_\sigma^{(l)}, \mathbf{v}_\sigma = c_\sigma^2 \cdot \mathbf{1}$, where $\mathbf{1}$ is a matrix or vector with all elements to be 1.

### 3.2 PAC-BAYESIAN LEARNING

Suppose data $S = \{(\mathbf{x}_i, y_i)\}_{i=1}^n$ are i.i.d. samples from a non-degenerate distribution $\mathcal{D}$. Define $\mathcal{H}$ to be the hypothesis space, $h(\mathbf{x})$ to be the prediction of hypothesis $h \in \mathcal{H}$ over for $\mathbf{x}$. Let

$R_{\mathcal{D}}(h) = \mathbb{E}_{(\mathbf{x},y)\sim\mathcal{D}}[\ell(y,h(\mathbf{x}))]$ represent the generalization error of classifier $h$ and $\widehat{R}_S(h) = \frac{1}{n}\sum_{i=1}^n \ell(y_i, h(\mathbf{x}_i))$ represent the empirical error of classifier $h$, where $\ell(\cdot)$ is the loss function.

In PAC-Bayes, the prior $Q(0) \in \mathcal{H}$ is the prior distribution in $\mathcal{H}$ at initialization or before training, and the posterior $Q \in \mathcal{H}$ is the distribution of parameters after training. To make the evaluation of prediction based on weight parameters $\mathbf{W}^{(l)}$ feasible for $l \in [L]$, we adopt the Gaussian distribution for the parameters, and the expected value for population risk and empirical error are $R_{\mathcal{D}}(Q) = \mathbb{E}_{(\mathbf{x},y)\sim\mathcal{D},h\sim Q}[\ell(y,h(\mathbf{x}))] = \mathbb{E}_{h\sim Q}[R_{\mathcal{D}}(h)]$, $\widehat{R}_S(Q) = \mathbb{E}_{h\sim Q}[\widehat{R}_S(h)]$. The PAC-Bayes theory (Langford & Seeger, 2001; Seeger, 2002; Maurer, 2004) gives the following theorem:

**Theorem 3.1.** *Then for any $\delta \in (0,1]$, the following inequality holds uniformly for all posteriors distributions $Q \in \mathcal{H}$ with a probability of at least $1 - \delta$,*

$$\mathrm{kl}\big(\widehat{R}_S(Q)\|R_{\mathcal{D}}(Q)\big) \leq \frac{\mathrm{KL}(Q\|Q(0)) + \log\frac{2\sqrt{n}}{\delta}}{n}. \tag{3}$$

where $\mathrm{KL}(Q\|Q(0)) = \mathbb{E}_Q\left[\ln\frac{Q}{Q(0)}\right]$ is the Kullback-Leibler (KL) divergence and $\mathrm{kl}(q\|q') = q\log(\frac{q}{q'}) + (1-q)\log(\frac{1-q}{1-q'})$ is the binary KL divergence. Furthermore, combined with Pinsker's inequality for binary KL divergence, $\mathrm{kl}(\hat{p}\|p) \geq (p-\hat{p})^2/(2p)$, when $\hat{p} < p$, yields,

$$R_{\mathcal{D}}(Q) - \widehat{R}_S(Q) \leq \sqrt{2R_{\mathcal{D}}(Q)\frac{\mathrm{KL}(Q\|Q(0)) + \log\frac{2\sqrt{n}}{\delta}}{n}}. \tag{4}$$

Equation (4) is a classical result. This result can be further combined with the inequality $\sqrt{ab} \leq \frac{1}{2}(\bar{\lambda}a + \frac{b}{\bar{\lambda}})$, for all $\bar{\lambda} > 0$, which leads to a PAC-Bayes-$\lambda$ bound in Theorem 3.2, as proposed by Thiemann et al. (2017):

**Theorem 3.2.** *Let $Q_0 \in \mathcal{H}$ be some prior distribution over $\mathcal{H}$. Then for any $\delta \in (0,1]$, the following inequality holds uniformly for all posteriors distributions $Q \in \mathcal{H}$ with a probability of at least $1 - \delta$*

$$R_{\mathcal{D}}(Q) \leq \frac{\widehat{R}_S(Q)}{1 - \bar{\lambda}/2} + \frac{\mathrm{KL}(Q\|Q(0)) + \log\frac{2\sqrt{n}}{\delta}}{n\bar{\lambda}(1 - \bar{\lambda}/2)}. \tag{5}$$

In this work, inspired by Catoni (2007); Rivasplata et al. (2019), we aim to promote the training objective as PAC-Bayes bound and choose Equation (5) as the training objective. We highlight that the original interest of Theorem 3.2 in (Thiemann et al., 2017) is to allow the optimization of a quasiconvex objective both $\bar{\lambda}$ and $Q$. However, since our main goal is to study the optimization and generalization properties of PNNs, we set directly $\bar{\lambda} = 1$ and we omit the factor of two and express the objective function as follows:

$$L(Q) = \widehat{R}_S(Q) + \lambda\frac{\mathrm{KL}(Q\|Q(0))}{n} = \mathbb{E}_{h\sim Q}\left[\frac{1}{n}\sum_{i=1}^n \ell(y_i, h(\mathbf{x}_i))\right] + \lambda\frac{\mathrm{KL}(Q\|Q(0))}{n} \tag{6}$$

where $\lambda$ is a hyperparameter introduced in a heuristic manner to make the method more flexible. Since the term regarding $\delta$ is a constant, we omit it in the objective function. We set $\ell$ to be the squared loss in the training objective function because it has a nice property such that the final solution of the output function is explicit in the infinite-width limit. The global convergence can be extended to cross-entropy loss like existing works (Ji & Telgarsky, 2019; Chen et al., 2019). Instead of optimizing $\mathbf{W}^{(l)}$ and $\mathbf{v}$ directly, the gradient descent with reparameterization trick leads to

$$\mathbf{W}_\mu^{(l)}(t+1) = \mathbf{W}_\mu^{(l)}(t) - \eta\frac{\partial L(Q)}{\partial\mathbf{W}_\mu^{(l)}(t)}; \quad \mathbf{W}_\sigma^{(l)}(t+1) = \mathbf{W}_\sigma^{(l)}(t) - \eta\frac{\partial L(Q)}{\partial\mathbf{W}_\sigma^{(l)}(t)} \tag{7}$$

where $\eta$ is the learning rate. For simplicity, we omit the gradient descent expression for $\mathbf{v}_\mu$ and $\mathbf{v}_\sigma$, and will omit the corresponding terms regarding $\mathbf{v}_\mu, \mathbf{v}_\sigma$ in the following text unless otherwise specified. To simplify theoretical analysis, this work considers gradient flow instead, and the same results can be extended to gradient descent case with a careful analysis.

## 4 MAIN THEORETICAL RESULTS

In this section, Theorem 4.2 gives a precise characterization of how the objective function without KL divergence decreases to zero. We then extend the convergence characterization to the full objective, and find the final solution is a kernel ridge regression, as demonstrated by Theorem 4.3. As a consequence, we are able to establish an analytic generalization bound through Theorem 4.4.

### 4.1 OPTIMIZATION ANALYSIS

To simplify the analysis, we first consider the optimization of probabilistic neural networks of the form (1) with objective $\widehat{R}_S(Q)$. In other words, we neglect the KL divergence term at this stage and show that the corresponding results can be extended to the target function with KL divergence in the next section. Given this premise, we show that for a $L$-layer probabilistic neural network, the gradient flow of output function admits the following dynamics

$$\frac{df(\mathbf{X};t)}{dt} = \frac{\partial f(\mathbf{X};t)}{\partial \boldsymbol{\theta}_\mu}\frac{\partial \boldsymbol{\theta}_\mu}{\partial t} + \frac{\partial f(\mathbf{X};t)}{\partial \boldsymbol{\theta}_\sigma}\frac{\partial \boldsymbol{\theta}_\sigma}{\partial t} = (\mathbf{y} - f(\mathbf{X};t))(\boldsymbol{\Theta}_\mu(\mathbf{X},\mathbf{X};t) + \boldsymbol{\Theta}_\sigma(\mathbf{X},\mathbf{X};t)) \quad (8)$$

where $\boldsymbol{\theta}_\mu \equiv (\{\mathbf{W}_\mu^{(l)}\}_{l=1}^L, \mathbf{v}_\mu)$ and $\boldsymbol{\theta}_\sigma \equiv (\{\mathbf{W}_\sigma^{(l)}\}_{l=1}^L, \mathbf{v}_\sigma)$ are collection of mean weights and variance weights. Besides, $\boldsymbol{\Theta}_\mu(\mathbf{X},\mathbf{X};t) \in \mathbb{R}^{n \times n}$ and $\boldsymbol{\Theta}_\sigma(\mathbf{X},\mathbf{X};t) \in \mathbb{R}^{n \times n}$ are *probabilistic neural tangent kernels* (PNTKs) defined as follows,

**Definition 4.1** (Probabilistic Neural Tangent Kernel). *The tangent kernels associated with the output function $f(\mathbf{X};t)$ at parameters $\boldsymbol{\theta}_\mu$ and $\boldsymbol{\theta}_\sigma$ are defined as,*

$$\boldsymbol{\Theta}_\mu(\mathbf{X},\mathbf{X};t) = \frac{\partial f(\mathbf{X};t)}{\partial \boldsymbol{\theta}_\mu}\left(\frac{\partial f(\mathbf{X};t)}{\partial \boldsymbol{\theta}_\mu}\right)^\top = \sum_{l=1}^L \nabla_{\mathbf{W}_\mu^{(l)}} f(\mathbf{X};t)\nabla_{\mathbf{W}_\mu^{(l)}} f(\mathbf{X};t)^\top + \nabla_{\mathbf{v}_\mu} f(\mathbf{X};t)\nabla_{\mathbf{v}_\mu} f(\mathbf{X};t)^\top$$

$$\boldsymbol{\Theta}_\sigma(\mathbf{X},\mathbf{X};t) = \frac{\partial f(\mathbf{X};t)}{\partial \boldsymbol{\theta}_\sigma}\left(\frac{\partial f(\mathbf{X};t)}{\partial \boldsymbol{\theta}_\sigma}\right)^\top = \sum_{l=1}^L \nabla_{\mathbf{W}_\sigma^{(l)}} f(\mathbf{X};t)\nabla_{\mathbf{W}_\sigma^{(l)}} f(\mathbf{X};t)^\top + \nabla_{\mathbf{v}_\sigma} f(\mathbf{X};t)\nabla_{\mathbf{v}_\sigma} f(\mathbf{X};t)^\top$$

$$(9)$$

Different from standard (deterministic) neural networks, the probabilistic network consist of two sets of parameters $\boldsymbol{\theta}_\mu$ and $\boldsymbol{\theta}_\sigma$, thus the PNTK has two corresponding tangent kernels.

One of the key findings of this work is that the PNTKs $\boldsymbol{\Theta}_\mu(\mathbf{X},\mathbf{X})$ and $\boldsymbol{\Theta}_\sigma(\mathbf{X},\mathbf{X})$ will both converge to a limiting *deterministic* kernel denoted as $\boldsymbol{\Theta}^\infty(\mathbf{X},\mathbf{X})$ at initialization and during training if $m$ is sufficiently large, namely $\lim_{m\to\infty} \boldsymbol{\Theta}_\mu(\mathbf{X},\mathbf{X}) = \boldsymbol{\Theta}^\infty(\mathbf{X},\mathbf{X})$ and $\lim_{m\to\infty} \boldsymbol{\Theta}_\sigma(\mathbf{X},\mathbf{X}) = \boldsymbol{\Theta}^\infty(\mathbf{X},\mathbf{X})$.

As a result, in the infinite-width limit, dynamics of output function with gradient flow is *linear*:

$$\frac{df(\mathbf{X};t)}{dt} = 2(\mathbf{y} - f(\mathbf{X};t))\boldsymbol{\Theta}^\infty(\mathbf{X},\mathbf{X}) \quad (10)$$

By leveraging this insight, we arrive at our main convergence theory for deep probabilistic neural networks, which is stated formally as follows

**Theorem 4.2** (Convergence of probabilistic networks with large width). *Suppose $\sigma(\cdot)$ is $H$-Lipschitz, $\lambda_0(\mathbf{K}_\infty) > 0$, and the network's width is of $m = \Omega\left(2^{O(L)}\max\left\{\frac{n^2\log(Ln/\delta)}{\lambda_0^2(\mathbf{K}_\infty^{(L)})}, \frac{n}{\delta}, \frac{n^5\log(2/\delta)^{10}}{\lambda_0^2(\mathbf{K}_\infty^{(L)})}\right\}\right)$ with the initialization. Then, with a probability of at least $1 - \delta$ over the random initialization, we have,*

$$\widehat{R}_S(Q(t)) \leq \exp\left(-\lambda_0(\mathbf{K}_\infty^{(L)})t\right)\widehat{R}_S(Q(0)) \quad (11)$$

*where we define $\mathbf{K}^{(l)}(\mathbf{x}_i,\mathbf{x}_j) \equiv (\mathbf{x}_i^{(l)})^\top \mathbf{x}_j^{(l)}$ and $\mathbf{K}_\infty^{(l)}(\mathbf{x}_i,\mathbf{x}_j) \equiv \lim_{m\to\infty} (\mathbf{x}_i^{(l)})^\top \mathbf{x}_j^{(l)}$*

Our theorem establishes that if $m$ is large enough, the expected training error converges to zero at a linear rate. In particular, the least eigenvalue of PNTK governs the convergence rate. Besides, we find the change of weight is bounded during training, which is consistent with the requirement of PAC-Bayes theory that the loss function is bounded.

## 4.2 Training with KL divergence

According to Equation (6), there is a KL divergence term in the objective function. We expand the KL-divergence for two Gaussian distributions, $\mathbb{P}(t) \equiv \mathcal{N}(\mu_t, \sigma_t^2)$, and $\mathbb{P}(0) \equiv \mathcal{N}(\mu_0, \sigma_0^2)$,

$$\mathrm{KL}(\mathbb{P}(t)|\mathbb{P}(0)) = \frac{1}{2}\left( \log \frac{\sigma_0}{\sigma_t} + \frac{(\mu_t - \mu_0)^2}{\sigma_0^2} + \frac{\sigma_t}{\sigma_0} - 1 \right) \tag{12}$$

We compare the gradient of mean and variance weights $\mathbf{w}_\mu^{(l)}$ $\mathbf{w}_\sigma^{(l)}$. With a direct calculation, we have $\frac{\partial f(\mathbf{x}_i)}{\partial \mathbf{w}_\mu^{(l)}} = \frac{1}{\sqrt{m}} \frac{\partial f(\mathbf{x}_i)}{\partial \mathbf{x}^{(l)}} \sigma'(\mathbf{w}^{(l)}\mathbf{x}^{(l-1)})\mathbf{x}^{(l-1)}$ and $\frac{\partial f(\mathbf{x}_i)}{\partial \mathbf{w}_\sigma^{(l)}} = \frac{1}{\sqrt{m}} \frac{\partial f(\mathbf{x}_i)}{\partial \mathbf{x}^{(l)}} \sigma'(\mathbf{w}^{(l)}\mathbf{x}^{(l-1)})\mathbf{x}^{(l-1)} \odot \boldsymbol{\xi}^{(l)}$. It is shown that there is one more random variable $\boldsymbol{\xi}^{(l)}$ associated with the gradient regarding variance weights, which results in the expected gradient norm being zero. Therefore, it is equivalent to fix $\mathbf{w}_\sigma$ during gradient descent training and we arrive at the conclusion that probabilistic neural network is performing kernel ridge regression in the infinite-width limit:

**Theorem 4.3.** *Consider gradient descent on objective function (6). Suppose $m \geq \mathrm{poly}(n, 1/\lambda_0, 1/\delta, 1/\mathcal{E})$. Then, with a probability of at least $1 - \delta$ over the random initialization, we have*

$$f\big(\mathbf{x}, Q(t)\big)|_{t=\infty} = \boldsymbol{\Theta}_\mu^\infty(\mathbf{x}, \mathbf{X})\big(\boldsymbol{\Theta}_\mu^\infty(\mathbf{X}, \mathbf{X}) + \lambda/c_\sigma^2 \mathbf{I}\big)^{-1}\mathbf{y} \pm \mathcal{E} \tag{13}$$

*where $f(\mathbf{x}, Q(t)) = \mathbb{E}_{f \sim Q(t)} f(\mathbf{x}; t)$ aligns with the definition of the empirical loss function.*

Theorem 4.3 reveals the regularization effect of the KL term in PAC-Bayesian learning, and presents an explicit expression for the convergence result of the output function.

## 4.3 Generalization analysis

We use squared loss to train the probabilistic neural network but adopt a general and suitable loss $\ell \in [0, 1]$ to evaluate the PNN's generalization. Recall that in Theorem 3.2, the PAC-Bayesian bound concerning the distribution at initialization and after optimization is given. Therefore, combined with the results from Theorem 4.3, we provide a generalization bound for PAC-Bayesian learning with ultra-wide condition.

**Theorem 4.4** (PAC-Bayesian bound with NTK). *Suppose data $S = \{(\mathbf{x}_i, y_i)\}_{i=1}^n$ are i.i.d. samples from a non-degenerate distribution $\mathcal{D}$, and $m \geq \mathrm{poly}(n, \lambda_0^{-1}, \delta^{-1})$. Consider any loss function $\ell : \mathbb{R} \times \mathbb{R} \to [0, 1]$ that is 1-Lipschitz in the first argument such that $\ell(y, y) = 0$. Then with a probability of at least $1 - \delta$ over the random initialization and the training samples, the probabilistic neural network (PNN) trained by gradient descent for $T \geq \Omega(\frac{1}{\eta\lambda_0} \log \frac{n}{\delta})$ iterations has population risk $R_\mathcal{D}(Q)$ that is bounded as follows:*

$$R_\mathcal{D}(Q) \leq \frac{\mathbf{y}^\top(\boldsymbol{\Theta}_\mu^\infty(\mathbf{X}, \mathbf{X}) + \lambda/c_\sigma^2 \mathbf{I})^{-1}\mathbf{y}}{nc_\sigma^2} + \frac{\lambda}{c_\sigma^2}\sqrt{\frac{\mathbf{y}^\top(\boldsymbol{\Theta}_\mu^\infty(\mathbf{X}, \mathbf{X}) + \lambda/c_\sigma^2 \mathbf{I})^{-2}\mathbf{y}}{n}} + O\left( \frac{\log \frac{2\sqrt{n}}{\delta}}{n} \right). \tag{14}$$

The proof can be found in the Appendix C. In this theorem, we establish a reasonable generalization bound for the PAC-Bayesian learning framework, thus providing a theoretical guarantee. Compared to the PAC-bayes bound (5), our bound is analytic and computable. We further demonstrate the advantage of PAC-Bayesian learning by comparing it with the Rademacher complexity-based generalization bound for deterministic neural networks with a kernel ridge solution.

**Theorem 4.5** (Rademacher bound with NTK). *Suppose data $S = \{(\mathbf{x}_i, y_i)\}_{i=1}^n$ are i.i.d. samples from a non-degenerate distribution $\mathcal{D}$, and $m \geq \mathrm{poly}(n, \lambda_0^{-1}, \delta^{-1})$. Consider any loss function $\ell : \mathbb{R} \times \mathbb{R} \to [0, 1]$ that is 1-Lipschitz in the first argument such that $\ell(y, y) = 0$. Then with a probability of at least $1 - \delta$ over the random initialization and training samples, the deterministic neural network trained by gradient descent for $T \geq \Omega(\frac{1}{\eta\lambda_0} \log \frac{n}{\delta})$ iterations has population risk $R_\mathcal{D}$ that is bounded as follows:*

$$R_\mathcal{D} \leq \sqrt{\frac{\mathbf{y}^\top(\boldsymbol{\Theta}_\mu^\infty(\mathbf{X}, \mathbf{X}) + \lambda/c_\sigma^2 \mathbf{I})^{-1}\mathbf{y}}{n}} + \frac{\lambda}{c_\sigma^2}\sqrt{\frac{\mathbf{y}^\top(\boldsymbol{\Theta}_\mu^\infty(\mathbf{X}, \mathbf{X}) + \lambda/c_\sigma^2 \mathbf{I})^{-2}\mathbf{y}}{n}} + O\left( \sqrt{\frac{\log \frac{n}{\lambda_0\delta}}{n}} \right). \tag{15}$$

Theorem 4.5 is obtained by following Theorem 5.1 in Hu et al. (2019), which presents a Rademacher complexity-based generalization bound for ultra-wide neural networks with a kernel ridge regression solution. Similar analysis for kernel regression without regularization based on NTK can be found in Arora et al. (2019a); Cao & Gu (2019).

The main difference between two generalization bounds is $\frac{\mathbf{y}^\top (\mathbf{\Theta}_\mu^\infty (\mathbf{X}, \mathbf{X}) + \lambda/c_\sigma^2 \mathbf{I})^{-1} \mathbf{y}}{n}$ versus $\sqrt{\frac{\mathbf{y}^\top (\mathbf{\Theta}_\mu^\infty (\mathbf{X}, \mathbf{X}) + \lambda/c_\sigma^2 \mathbf{I})^{-1} \mathbf{y}}{n}}$, which is due to the fact that the PAC-Bayesian bound count the KL divergence while Rademacher bound calculate the reproducing kernel Hilbert space (RKHS) norm. We find that the convergence rate of the focused term are different. One is $O(1/n)$ and the other is $O(1/\sqrt{n})$. Therefore, we conclude that the PAC-Bayesian bound has a numerical improvement over the Rademacher complexity-based bound when $n$ is large.

## 5 PROOF SKETCH

To prove Theorem 4.2, we first show that PNTKs at initialization are close to the limiting kernel given the width is large enough. Then we prove the distance between PNTKs and limiting kernel during training is also bounded, meanwhile loss has a linear convergence rate by induction. Our proof framework is similar to Du et al. (2019); Arora et al. (2019a)'s. However, The main difference is that our network architecture is much more complex (e.g. probabilistic network contains two sets of parameters) and each set involves its own randomness which requires bounding many terms more elaborately. The detailed proof can be found in Appendix A.

The proof of Theorem 4.3 utilizes an argument of linearization of the network model in the infinite-width limit. This allows us to obtain an ordinary differential equation for output function with the solution of kernel ridge regression. The details are given in Appendix B.

For generalization analysis, we defer the proofs of Theorem 4.4 to Appendix C. Our proof is based on a characterization of the empirical error and KL divergence term via the explicit solution found in Theorem 4.3.

## 6 EXPERIMENTS

As an extension of our finding of the PAC-Bayesian bound in Theorem 4.4, we provide a training-free metric to approximate the PAC-Bayesian bound via PNTK, which can be used to select the best hyperparameters without involving any training and eliminate excessive computation time. Besides, we provide a empirical verification of our theory in Appendix D.1 and comparison of theoretical bounds with empirical bounds in Appendix D.2.

### 6.1 EXPERIMENTAL SETUP

In all experiments, the NTK parameterization is chosen to initialize the parameters, which follows Equation (1). Specifically, the initial mean weights $\boldsymbol{\theta}_\mu$, are sampled from a truncated Gaussian distribution with a mean of zero and one standard variance of 1, truncating at two standard deviations. To ensure that the variance is positive, the initial variance for weight is transformed from the given value of $c_\sigma$ through the formula $c_\sigma = \log(1 + \exp(\rho_0))$.

In section 6.2, we describe the use of both fully connected and convoluted neural network structures to perform experiments on MINIST and CIFAR10 datasets to demonstrate the effectiveness of our training-free PAC-Bayesian network bound for searching hyperparameters under different datasets and network structures. In particular, we build a 3-layer fully-connected neural network with 600 neurons on each layer. On the other hand, the convolutional architecture is equipped with a total of 13 layers with around 10 million learnable parameters. We adopt a data-dependent prior since it is a practical and popular method (Perez-Ortiz et al., 2021; Fortuin, 2022). Specifically, this data-dependent prior is pre-trained on a subset of total training data with empirical risk minimization. The networks for posterior training are then initialized by the weights learned from the prior. Finally, the generalization bound is computed using Equation (5). The relevant settings are referred to in the work by Pérez-Ortiz et al. (2020), such as confidence parameters for the risk certificate and Chernoff bound, and the 150,000 times of Monte Carlo samples to estimate the risk certificate.

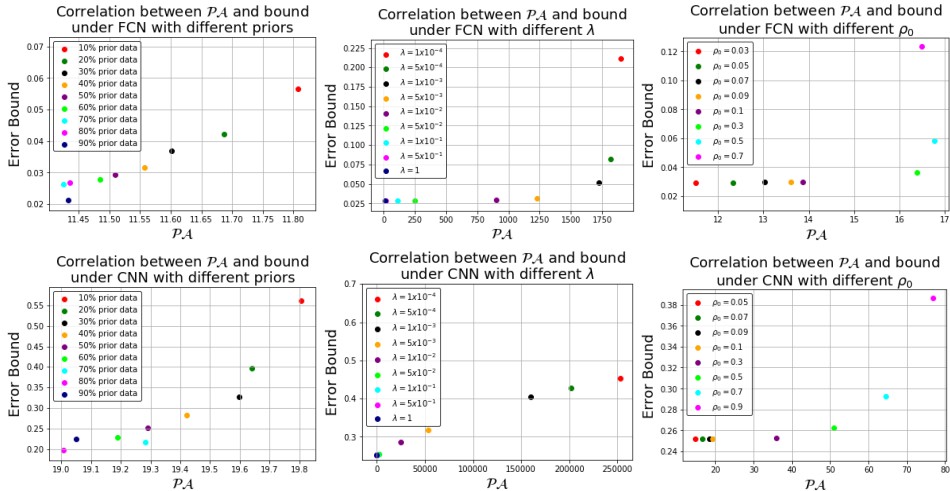

Figure 1: The first row shows correlation results of FCN structure on the MNIST dataset. Kendall-tau correlations between generalization bound with respect to the proportion of prior data, coefficient of KL penalty, and $\rho_0$ are 0.89, 0.89, and 0.93 at 1% level of significance. Similar results are found in the CNN structure with the CIFAR10 dataset where Kendall-tau correlations are 0.89, 0.83, and 0.57, as shown in the second row.

| Setup | | | Risk cert. | | Computation time (hours) | |
|---|---|---|---|---|---|---|
| Data | Method | Network | $\ell^{\text{x-e}}$ | $\ell^{01}$ | Single | Total |
| MNIST | Exhaustive Search | FCN | .0010 | .0212 | 0.50 | 324.00 |
| | | CNN | **.0059** | **.0110** | 16.92 | 10964.16 |
| | Bayesian Search | FCN | .0010 | .0212 | 0.50 | **18.00** |
| | | CNN | **.0059** | **.0110** | 16.92 | 609.12 |
| | $\mathcal{PA}$ | FCN | .0010 | .0264 | 0.03 | 19.44 |
| | | CNN | .0085 | .0160 | 0.03 | 19.44 |
| CIFAR10 | Exhaustive Search | FCN | .174 | 0.5377 | 1.09 | 706.32 |
| | | CNN | **.0142** | **.1969** | 45.00 | 29,160.00 |
| | Bayesian Search | FCN | .174 | 0.5377 | 1.09 | 39.24 |
| | | CNN | **.0142** | **.1969** | 45.00 | 1,620.00 |
| | $\mathcal{PA}$ | FCN | .178 | 0.5490 | 0.03 | **19.44** |
| | | CNN | **.0142** | .1970 | 0.03 | **19.44** |

Table 1: The performance i.e., risk certificates (cross-entropy $\ell^{\text{x-e}}$ and accuracy $\ell^{01}$) and computation time against three hyperparameters searching methods (exhaustive search, Bayesian search, and $\mathcal{PA}$, the training free method). For the lowest risk certificate and computational time are highlighted in boldface, and second best are highlighted by underlining.

## 6.2 SELECTING HYPERPARAMETERS VIA TRAINING-FREE METRIC

The PAC-Bayesian learning framework provides competitive performance with non-vacuous generalization bounds. However, the tightness of this generalization bound depends on the hyperparameters used, such as the proportionality of data used for the prior, the initialization of $\rho_0$, and the KL penalty weight ($\lambda$). Since these three values do not change during the training, we refer to them as hyperparameters. Choosing the right hyperparameters via a grid search is obviously prohibitive, as each attempt to compute the generalization bound can involve significant computational resources.

Another plausible approach is to design some kind of predictive, "training-free" metric so that we can approximate the error bound without going through an expensive training process. In light of this goal, we have already developed a generalization bound in theorem 4.4 via NTK. Since NTK changes are held constant during training, we can predict the generalization bound by this proxy metric, which can be formulated as follows:

$$\mathcal{PA} = \mathrm{Tr} \left( \frac{(\widehat{\mathbf{\Theta}} + \lambda/c_\sigma^2 \mathbf{I})^{-1} \cdot \mathbf{y}\mathbf{y}^\top}{c_\sigma^2 \cdot n} + \frac{\lambda}{c_\sigma^2} \sqrt{\frac{(\widehat{\mathbf{\Theta}} + \lambda/c_\sigma^2 \mathbf{I})^{-2} \cdot \mathbf{y}\mathbf{y}^\top}{n}} \right) \qquad (16)$$

where $\widehat{\mathbf{\Theta}}$ is an empirical NTK associated with mean weights, measured on a finite-width neural network at initialization. $\mathbf{y}\mathbf{y}^\top$ is a $n \times n$ label similarity matrix (if two data have the same label, their joint entry in the matrix is one and zero otherwise), and $n$ is the number of data used. Note that the proposed proxy metric in Eq. (16) share the same spirit of kernel alignment, a label similarity metric, which has been widely used in the application of deep active learning (Wang et al., 2021), model selection for fine-tuning (Deshpande et al., 2021), and neural architecture search (NAS) (Mok et al., 2022). To demonstrate the computational practicality of this training-free metric, we compute $\mathcal{PA}$ using only a subset of the data for each class (325 per class for FCN and 75 per class for CNN). We should also mention that training-free methods for searching neural architectures are not new, and can be found in NAS (Chen et al., 2021; Deshpande et al., 2021), MAE Random Sampling (Camero et al., 2021), pruning at initialization (Abdelfattah et al., 2021). To the best of our knowledge, there is currently no training-free method for selecting hyperparameters in the PAC-Bayesian framework, which we consider to be one of the novelties of this paper.

Figure 1 demonstrates a strong correlation between $\mathcal{PA}$ and the actual generalization bound. Finally, we demonstrate that by searching through all possible combinations of hyperparameters using $\mathcal{PA}$, it is possible to select a hyperparameter leading towards a result that is comparable to the best generalization bound, but without excessive computation. To put things in perspective, in Table 1, we compare the risk certificates and computation time for three hyperparameters finding methods (exhaustive search, Bayesian search and $\mathcal{PA}$) on the two architectures (FCN and CNN) and two datasets (MNIST and CIFAR10). Unlike exhaustively searching where the best set of hyperparameters are selected from 648 different hyperparameter combinations (9 data-dependent prior with different subsets data for prior training, 9 different values of KL penalty, and 8 different values of $\rho_0$), Bayesian search takes only 36 iterations to find the lowest bound since it evaluates the information in past iterations of searching and efficiently selecting the next set of hyperparameters based on the prior knowledge. Yet, reducing the number of search iterations cannot sufficiently reduce the overall computation time when training a large and complex model. For instance, under the CIFAR10 dataset, it takes 45 hours to train a CNN with the bound. In contrast, using the training-free method of $\mathcal{PA}$ save 83.33 times the computational time to find the bound that is close to the lowest risk certificate in accuracy.

## 7 DISCUSSION

In this work, we theoretically prove that the learning dynamics of deep probabilistic neural networks using training objectives derived from PAC-Bayes bounds are exactly described by the NTK in an over-parameterized setting. Empirical investigation reveals that this agrees well with the actual training process. Furthermore, the expected output function trained with a PAC-Bayesian bound converges to the kernel ridge regression under a mild assumption. Based on this finding, we obtain an explicit generalization bound with respect to NTK for PAC-Bayesian learning, which improves over the generalization bound obtained through NTK on a non-probabilistic neural network. Finally, we show that the PAC-Bayesian bound score, the training-free method, can effectively select the hyperparameters which leads to a lower generalization bound without cost excessive computation time cost which the brute-force grid search would incur. In summary, we establish our theoretical analysis on PAC-Bayes with a random initialized prior. Notice that neural tangent kernel cannot characterize the feature learning process in deep learning (Damian et al., 2022; Ba et al., 2022). This paper does not try to capture the feature learning for probabilistic neural networks given the NTK techniques used, but does provide sufficient new important convergence and generalization analysis for PAC-Bayesian learning. One promising direction would be to study PAC-Bayesian learning with data-dependent priors by NTK.

## 8 REPRODUCIBILITY STATEMENT

To ensure the results and conclusions of our paper are reproducible, we make the following efforts:

Theoretically, we state the full set of assumptions and include complete proofs of our theoretical results in Section 4; Appendix A, B, and C.

Experimentally, we provide our code, and instructions needed to reproduce the main experimental results. And we specify all the training and implementation details in Section 6 and Appendix D.

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

# A    PROOF OF THEOREM 4.2

**Theorem A.1** (Restatement of Theorem 4.2). *Suppose $\sigma(\cdot)$ is $H$-Lipschitz and the network's width is of $m = \Omega\left(2^{O(L)}\max\left\{\frac{n^2\log(Ln/\delta)}{\lambda_0^2(\mathbf{K}_\infty^{(L)})}, \frac{n}{\delta}, \frac{n^5\log(2/\delta)^{10}}{\lambda_0^2(\mathbf{K}_\infty^{(L)})}\right\}\right)$ with the initialization. Then, with a probability of at least $1-\delta$ over the random initialization, we have,*

$$\widehat{R}_S(Q(t)) \leq \exp\left(-\lambda_0(\mathbf{K}_\infty^{(L)})t\right)\widehat{R}_S(Q(0))$$

*where we define $\mathbf{K}^{(l)}(\mathbf{x}_i, \mathbf{x}_j) \equiv (\mathbf{x}_i^{(l)})^\top \mathbf{x}_j^{(l)}$ and $\mathbf{K}_\infty^{(l)}(\mathbf{x}_i, \mathbf{x}_j) \equiv \lim_{m\to\infty}(\mathbf{x}_i^{(l)})^\top \mathbf{x}_j^{(l)}$*

*Proof Sketch of Theorem A.1.* To study the behavior of output function under gradient flow, we first write down its dynamics

$$\frac{df(\mathbf{X};t)}{dt} = \frac{\partial f(\mathbf{X};t)}{\partial \boldsymbol{\theta}_\mu}\frac{\partial \boldsymbol{\theta}_\mu}{\partial t} + \frac{\partial f(\mathbf{X};t)}{\partial \boldsymbol{\theta}_\sigma}\frac{\partial \boldsymbol{\theta}_\sigma}{\partial t} = (\mathbf{y} - f(\mathbf{X};t))(\boldsymbol{\Theta}^\mu(\mathbf{X},\mathbf{X};t) + \boldsymbol{\Theta}^\sigma(\mathbf{X},\mathbf{X};t))$$

where $\boldsymbol{\Theta}^\mu$ and $\boldsymbol{\Theta}^\sigma$ are the PNTKs of the whole network, composed by the NTK of each layer. We observe that if $\boldsymbol{\Theta}^\mu$ and $\boldsymbol{\Theta}^\sigma$ converge to a deterministic kernel, then the dynamics of output function admit a linear system, which is tractable during evolution.

Before demonstrating the main steps, we introduce a Neural Network Gaussian Process (NNGP) of our studied neural network in the infinite-width limit (Lee et al., 2017), which is defined as follows:

$$\mathbf{K}^{(l)}(\mathbf{x}_i, \mathbf{x}_j) \equiv (\mathbf{x}_i^{(l)})^\top \mathbf{x}_j^{(l)} \quad \mathbf{K}_\infty^{(l)}(\mathbf{x}_i, \mathbf{x}_j) \equiv \lim_{m\to\infty}(\mathbf{x}_i^{(l)})^\top \mathbf{x}_j^{(l)}$$

where subscript $i, j$ denote the index of input samples. Instead of showing the $\boldsymbol{\Theta}^\mu$ and $\boldsymbol{\Theta}^\sigma$ are close to $\boldsymbol{\Theta}^\infty$ in infinite-width limit, we use $\mathbf{K}^{(L)}$ as an anchor kernel. With the relation of NKT and NNGP, we can simplify our proof. Therefore, to prove Theorem A.1, three core steps are:

Step 1   Show at initialization $\lambda_{\min}(\boldsymbol{\Theta}_\mu(0)), \lambda_{\min}(\boldsymbol{\Theta}_\sigma(0)) \geq \frac{\lambda_{\min}(\mathbf{K}^{(L)})}{2}$ and the required condition on $m$.

Step 2   Show during training $\lambda_{\min}(\boldsymbol{\Theta}_\mu(0)), \lambda_{\min}(\boldsymbol{\Theta}_\sigma(0)) \geq \frac{\lambda_{\min}(\mathbf{K}^{(L)})}{2}$ and the required condition on $m$.

Step 3   Show during training the empirical loss has a linear convergence rate.

In our proof, we mainly focus on deriving the condition on $m$ by analyzing $\lambda_{\min}(\boldsymbol{\Theta}_\mu(0))$ and $\lambda_{\min}(\boldsymbol{\Theta}_\sigma(0))$ at initialization through Lemma A.2 and Lemma A.3. For step 2, we construct Lemma A.5 and Lemma A.6 to demonstrate that $\lambda_{\min}(\boldsymbol{\Theta}_\mu(0)), \lambda_{\min}(\boldsymbol{\Theta}_\sigma(0)) \geq \frac{\lambda_{\min}(\mathbf{K}^{(L)})}{2}$. This This leads to the conclusion that the required condition on $m$ during train. Finally, we summarize all the previous lemmas and conclude that the training error converges at a linear rate through Lemma A.7.

$\square$

## A.1    STEP 1. BOUNDING LEAST EIGENVALUE OF PNTK AT INITIALIZATION

We first study the behavior of tangent kernels with an ultra-wide condition, namely $m = \text{poly}(n, 1/\lambda_0, 1/\delta)$ at initialization. Lemmas A.2 and A.3 demonstrate that if $m$ is large, then the feature of each layer is approximately normalized, $\boldsymbol{\Theta}^\mu(0)$ and $\boldsymbol{\Theta}^\sigma(0)$ have a lower bound on the smallest eigenvalue with a high probability.

**Lemma A.2** (Initial norm at initialization). *Suppose $\sigma(\cdot)$ is $H$-Lipschitz. If $m = \Omega\left(\frac{nLg_C(L)^2}{\delta}\right)$, where $C \equiv (c_\mu^2 + c_\sigma^2)H(2|\sigma(0)|\sqrt{\frac{2}{\pi}} + 2H)$, while $\mathbf{W}_\mu$ and $\mathbf{W}_\sigma$ are initialized by the form described in Section 3.1, then with probability at least $1-\delta$ over random initialization, for each $l \in [L]$ and $i \in [n]$, we have*

$$\frac{1}{2} \leq \|\mathbf{x}_i^{(l)}(0)\|_2 \leq 2$$

*where the geometric series function $g_C(l) = \sum_{i=0}^{n-1} C^i$.*

**Lemma A.3** (PNTK at initialization). *Suppose $\sigma(\cdot)$ is $H$-Lipschitz. If $m = \Omega\left(\frac{n^2 \log(Ln/\delta) 2^{O(L)}}{\lambda_{\min}^2(\mathbf{K}^{(L)})}\right)$, while $\mathbf{W}_\mu$ and $\mathbf{W}_\sigma$ are initialized by the form described in Section 3.1, then with probability at least $1 - \delta$, we have*

$$\lambda_{\min}(\mathbf{\Theta}_\mu^{(L)}(0)) \geq \frac{3}{4}\lambda_{\min}(\mathbf{K}^{(L)})$$

$$\lambda_{\min}(\mathbf{\Theta}_\sigma^{(L)}(0)) \geq \frac{3}{4}\lambda_{\min}(\mathbf{K}^{(L)})$$

*Proof of Lemma A.2.* The proof is by induction method. The induction hypothesis is that with probability at least $1 - (l-1)\frac{\delta}{nL}$ over $\mathbf{W}^{(1)}(0), \dots, \mathbf{W}^{(l-1)}(0)$, for every $1 \leq l' \leq l-1$, we have

$$\frac{1}{2} \leq 1 - \frac{g_C(l')}{2g_C(L)} \leq \|\mathbf{x}_i^{(l')}(0)\|_2 \leq 1 + \frac{g_C(l')}{2g_C(L)} \leq 2$$

where the geometric series function $g_C(l) = \sum_{i=0}^{n-1} C^i$.

Note that there are two randomness in each $\mathbf{W}^{(l)}$ for $l \in [L]$, which can be seen from the expression:

$$\mathbf{W}^{(l)} = \mathbf{W}_\mu^{(l)} + \mathbf{W}_\sigma^{(l)} \odot \boldsymbol{\xi}^{(l)}$$

The first randomness comes from the initialization of $\mathbf{W}_\mu^{(l)}$, and the second randomness is from random variable $\boldsymbol{\xi}$. We then unify the two randomness into one, namely $\mathbf{W}^{(l)} \sim \mathcal{N}(\mathbf{0}, (c_\mu^2 + c_\sigma^2) \cdot \mathbf{I})$ through the following argument:

$$\mathbb{P}(W_{ij}^{(l)}) = \frac{1}{\sqrt{2\pi}c_\sigma} e^{-\frac{\left(W_{ij}^{(l)} - W_{\mu,ij}^{(l)}\right)^2}{2c_\sigma^2}} \mathbb{P}(W_{\mu,ij}^{(l)})$$

Plugging the density function $\mathbb{P}(W_{\mu,ij})$ of variable $\boldsymbol{\mu}_i$ into the above expression, we can obtain,

$$\mathbb{P}(W_{ij}^{(l)}) = \int \frac{1}{\sqrt{2\pi}c_\sigma} e^{-\frac{\left(W_{ij}^{(l)} - W_{\mu,ij}^{(l)}\right)^2}{2c_\sigma^2}} \frac{1}{\sqrt{2\pi}c_\mu} e^{-\frac{\left(W_{\mu,ij}^{(l)} - 0\right)^2}{2c_\mu^2}} dW_{\mu,ij}^{(l)}$$

$$= \frac{1}{\sqrt{2\pi(c_\mu^2 + c_\sigma^2)}} e^{-\frac{\left(W_{ij}^{(l)} - 0\right)^2}{2(c_\mu^2 + c_\sigma^2)}}$$

With the result of $\mathbb{P}(W_{\mu,ij})$ at hand, we continue to bound $\|\mathbf{x}_i^{(l)}\|_2^2$. We calculate the expectation over the randomness from $\mathbf{W}^{(l)}(0)$. According to the feed-forward expression, we know that

$$\|\mathbf{x}_i^{(l)}(0)\|_2^2 = \frac{1}{m} \sum_{r=1}^{m} \sigma(\mathbf{w}_r^{(l)}(0)^\top \mathbf{x}_i^{l-1}(0))^2$$

Then we have

$$\mathbb{E}\left[\|\mathbf{x}_i^{(l)}(0)\|_2^2\right] = \mathbb{E}\left[\sigma(\mathbf{w}_r^{(l)}(0)^\top \mathbf{x}_i^{(l-1)}(0))^2\right]$$

$$= (c_\mu^2 + c_\sigma^2)\mathbb{E}_{Z \sim \mathcal{N}(0,1)}\sigma(\|\mathbf{x}^{(l-1)}\|_2 Z)^2$$

Because $\sigma(\cdot)$ is $H$-Lipschitz, for $\frac{1}{2} \le \alpha \le 2$, we have

$$
\begin{aligned}
&\left| \mathbb{E}_{Z \sim \mathcal{N}(0,1)} \left[ \sigma(\alpha Z)^2 \right] - \mathbb{E}_{Z \sim \mathcal{N}(0,1)} \left[ \sigma(Z)^2 \right] \right| \\
&\le \mathbb{E}_{Z \sim \mathcal{N}(0,1)} \left[ \left| \sigma(\alpha Z)^2 - \sigma(Z)^2 \right| \right] \\
&\le H|\alpha - 1| \cdot \mathbb{E}_{Z \sim \mathcal{N}(0,1)} \left[ |Z(\sigma(\alpha Z) + \sigma(Z))| \right] \\
&\le H|\alpha - 1| \cdot \mathbb{E}_{Z \sim \mathcal{N}(0,1)} \left[ |Z|(|2\sigma(0)| + H|(\alpha + 1)Z|) \right] \\
&\le H|\alpha - 1| \cdot (2|\sigma(0)| \mathbb{E}_{Z \sim \mathcal{N}(0,1)}[|Z|] + H|\alpha + 1| \mathbb{E}_{Z \sim \mathcal{N}(0,1)}[Z^2]) \\
&= H|\alpha - 1| \cdot (2|\sigma(0)| \sqrt{\frac{2}{\pi}} + H|\alpha + 1|) \\
&\le \frac{C}{c_\mu^2 + c_\sigma^2} |\alpha - 1|
\end{aligned}
$$

where we define $C \equiv (c_\mu^2 + c_\sigma^2) H (2\sigma(0)\sqrt{\frac{2}{\pi}} + 2H)$.

For the variance we have

$$
\begin{aligned}
\mathrm{Var}\left[ \|\mathbf{x}_i^{(l)}(0)\|_2^2 \right] &= \frac{(c_\sigma^2 + c_\mu^2)^2}{m} \mathrm{Var}\left[ \sigma \mathbf{w}_r^{(l)}(0)^\top \mathbf{x}_i^{(l)}(0)^2 \right] \\
&\le \frac{(c_\sigma^2 + c_\mu^2)^2}{m} \mathbb{E}\left[ \sigma(\mathbf{w}_r^{(l)}(0)^\top \mathbf{x}_i^{(l)}(0))^4 \right] \\
&\le \frac{(c_\sigma^2 + c_\mu^2)^2}{m} \mathbb{E}\left[ \left( |\sigma(0)| + H|\mathbf{w}_r^{(l)}(0)^\top \mathbf{x}_i^{(l)}(0)| \right)^4 \right] \\
&\le \frac{C_2}{m}.
\end{aligned}
$$

where $C_2 \equiv \sigma(0)^4 + 8|\sigma(0)|^3 H \sqrt{2/\pi} + 24\sigma(0)^2 H^2 + 64\sigma(0) H^3 \sqrt{2/\pi} + 512 H^4$ and the last inequality we used the formula for the first four absolute moments of Gaussian.

Applying Chebyshev's inequality and plugging in our assumption on $m$, we have with probability $1 - \frac{\delta}{nL}$ over $\mathbf{W}^{(l)}$,

$$
\left| \|\mathbf{x}_i^{(l)}(0)\|_2^2 - \mathbb{E}\|\mathbf{x}_i^{(l)}(0)\|_2^2 \right| \le \frac{1}{2g_C(L)}.
$$

Thus with probability $1 - d\frac{\delta}{nL}$ over $\mathbf{W}^{(1)}, \ldots, \mathbf{W}^{(l)}$,

$$
\begin{aligned}
\left| \|\mathbf{x}_i^{(l)}(0)\|_2 - 1 \right| &\le \left| \|\mathbf{x}_i^{(l)}(0)\|_2^2 - 1 \right| \\
&\le \frac{C g_C(l-1)}{2g_C(L)} + \frac{1}{2g(L)} \\
&= \frac{g_C(l)}{2g_C(L)}
\end{aligned}
$$

Using union bounds over $[n]$, we prove the lemma.

$\square$

*Proof of Lemma A.3.* For a weight matrix, we decompose it into $m$ weight vectors, namely $\mathbf{W}^{(l)} = [\mathbf{w}_1^{(l)}, \mathbf{w}_2^{(l)}, \cdots, \mathbf{w}_m^{(l)}]$. Then the derivative of output over the parameters $\mathbf{w}_{\mu,r}^{(l)}$ and $\mathbf{w}_{\sigma,r}^{(l)}$ can be expressed as

$$
\begin{aligned}
\frac{\partial f(\mathbf{x}_i)}{\partial \mathbf{w}_{\mu,r}^{(l)}} &= \frac{1}{\sqrt{m}} \frac{\partial f(\mathbf{x}_i)}{\partial \mathbf{x}^{(l)}} \sigma'(\mathbf{w}_r^{(l)} \mathbf{x}^{(l-1)}) \mathbf{x}^{(l-1)} \\
\frac{\partial f(\mathbf{x}_i)}{\partial \mathbf{w}_{\sigma,r}^{(l)}} &= \frac{1}{\sqrt{m}} \frac{\partial f(\mathbf{x}_i)}{\partial \mathbf{x}^{(l)}} \sigma'(\mathbf{w}_r^{(l)} \mathbf{x}^{(l-1)}) \mathbf{x}^{(l-1)} \odot \boldsymbol{\xi}_r^{(l)}
\end{aligned}
\tag{17}
$$

According to the definition of PNTK for each layer:

$$\mathbf{\Theta}_\mu^{(l)} = \nabla_{\mathbf{W}_\mu^{(l)}} f(\mathbf{X}; t) \nabla_{\mathbf{W}_\mu^{(l)}} f(\mathbf{X}; t)^\top$$

$$\mathbf{\Theta}_\sigma^{(l)} = \nabla_{\mathbf{W}_\sigma^{(l)}} f(\mathbf{X}; t) \nabla_{\mathbf{W}_\sigma^{(l)}} f(\mathbf{X}; t)^\top$$

Through a standard calculation, we show that PNTKs can be expressed as

$$\mathbf{\Theta}_{\mu,ij}^{(L)} = (\mathbf{x}_i^{(L-1)})^\top \mathbf{x}_j^{(L-1)} \cdot \frac{1}{m} \sum_{r=1}^m v_r^2 \sigma'((\mathbf{w}_r^{(L)})^\top \mathbf{x}_i^{(L-1)}) \sigma'((\mathbf{w}_r^{(L)})^\top \mathbf{x}_j^{(L-1)})$$

$$\mathbf{\Theta}_{\sigma,ij}^{(L)} = (\mathbf{x}_i^{(L-1)})^\top \mathbf{x}_j^{(L-1)} \cdot \frac{1}{m} \sum_{r=1}^m v_r^2 \sigma'((\mathbf{w}_r^{(L)})^\top \mathbf{x}_i^{(L-1)}) \sigma'((\mathbf{w}_r^{(L)})^\top \mathbf{x}_j^{(L-1)}) \cdot \xi_r^2$$

Note that the difference between $\mathbf{\Theta}_\mu^{(L)}$, $\mathbf{\Theta}_\sigma^{(L)}$ and $\mathbf{K}^{(L)}$ can be decomposed as follows:

$$\mathbf{\Theta}_\mu^{(L)} - \mathbf{K}^{(H)} = (\mathbf{\Theta}_\mu^{(L)}(0) - \mathbf{K}_\infty^{(L)}) + (\mathbf{K}_\infty^{(L)} - \mathbf{K}^{(L)})$$

$$\mathbf{\Theta}_\sigma^{(L)} - \mathbf{K}^{(H)} = (\mathbf{\Theta}_\sigma^{(L)}(0) - \mathbf{K}_\infty^{(L)}) + (\mathbf{K}_\infty^{(L)} - \mathbf{K}^{(L)})$$

We split the proof process into two phases:

- First we use concentration inequality to show that if $m = \Omega\left(\frac{n^2 \log(n^2/\delta)}{\lambda_{\min}^2(\mathbf{K}^{(L)})}\right)$, we have

$$\left\|\mathbf{\Theta}_\mu^{(L)}(0) - \mathbf{K}_\infty^{(L)}\right\|_2 \leq \frac{\lambda_{\min}(\mathbf{K}^{(L)})}{4} \quad \left\|\mathbf{\Theta}_\sigma^{(L)}(0) - \mathbf{K}_\infty^{(L)}\right\|_2 \leq \frac{\lambda_{\min}(\mathbf{K}^{(L)})}{4}$$

- Second, we show that if $m = \Omega\left(\frac{n^2 \log(Ln/\delta)2^{O(L)}}{\lambda_{\min}^2(\mathbf{K}^{(L)})}\right)$, then we have,

$$\left\|\mathbf{K}_\infty^{(L)} - \mathbf{K}^{(L)}\right\|_\infty \leq \frac{\lambda_{\min}(\mathbf{K}^{(L)})}{2}$$

**Phase 1, bounding $\mathbf{\Theta}_\mu^{(L)}$.** Plugging the derivative result regarding mean weights in Equation (17) into the definition of PNTK (Eqution 9) yields:

$$\mathbf{\Theta}_{\mu,ij}^{(L)}(0) = (\mathbf{x}_i^{(L-1)})^\top \mathbf{x}_j^{(L-1)} \cdot \frac{1}{m} \sum_{r=1}^m v_r^2 \sigma'((\mathbf{w}_r^{(L)})^\top \mathbf{x}_i^{(L-1)}) \sigma'((\mathbf{w}_r^{(L)})^\top \mathbf{x}_j^{(L-1)})$$

By an analysis, we find that for all pairs of $i, j$, $\mathbf{\Theta}_{\mu,ij}^{(L)}(0)$ is the average of $m$ i.i.d. random variables, with the expectation

$$\mathbf{K}_{\infty,ij}^{(L)} = (c_\mu^2 + c_\sigma^2) \cdot \mathbb{E}_{\mathbf{w} \sim \mathcal{N}(\mathbf{0}, \mathbf{I})} \left[ (\mathbf{x}_i^{(L-1)})^\top \mathbf{x}_j^{(L-1)} \sigma'(\mathbf{w}^\top \mathbf{x}_i^{(L-1)})^\top \sigma'(\mathbf{w}^\top \mathbf{x}_j^{(L-1)}) \right]$$

Then by Hoeffding's inequality, we know that the following inequality holds with probability at least $1 - \delta'$,

$$\left| \mathbf{\Theta}_{\mu,ij}^{(L)}(0) - \mathbf{K}_{\infty,ij}^{(L)} \right| \leq \sqrt{\frac{\log(2/\delta')}{2m}}$$

Because NTK matrix is of size $n \times n$, we then apply a union bound over all $i, j \in [n]$ (by setting $\delta' = \delta/n^2$), and obtain that

$$\left| \mathbf{\Theta}_{\mu,ij}^{(L)}(0) - \mathbf{K}_{\infty,ij}^{(L)} \right| \leq \sqrt{\frac{\log(2n^2/\delta)}{2m}}$$

Thus we have,

$$
\begin{aligned}
\left\| \mathbf{\Theta}_\mu^{(L)}(0) - \mathbf{K}_\infty^{(L)} \right\|_2^2 &\leq \left\| \mathbf{\Theta}_\mu^{(L)}(0) - \mathbf{K}_\infty^{(L)} \right\|_F^2 \\
&\leq \sum_{i,j} \left| \mathbf{\Theta}_{\mu,ij}^{(L)}(0) - \mathbf{K}_{\infty,ij}^{(L)} \right|^2 \\
&= O\left( \frac{n^2 \log(2n^2/\delta)}{m} \right)
\end{aligned}
$$

Finally, if $\sqrt{\frac{n^2 \log(2n^2/\delta)}{m}} \leq \frac{\lambda_{\min}(\mathbf{K}^{(L)})}{4}$, which implies $m = \Omega\left( \frac{n^2 \log(n^2/\delta)}{\lambda_{\min}^2(\mathbf{K}^{(L)})} \right)$, then with probability at least $1 - \delta$,

$$
\left\| \mathbf{\Theta}_\mu^{(L)}(0) - \mathbf{K}_\infty^{(L)} \right\|_2 \leq \frac{\lambda_{\min}(\mathbf{K}^{(L)})}{4}
$$

**Phase 1, bounding $\mathbf{\Theta}_\sigma^{(L)}(0)$.** Plugging the derivative result regarding mean weights in Equation (17) into the definition of PNTK (Eqution 9) yields:

$$
\mathbf{\Theta}_{\sigma,ij}^{(L)}(0) = (\mathbf{x}_i^{(L-1)})^\top \mathbf{x}_j^{(L-1)} \cdot \frac{1}{m} \sum_{r=1}^m v_r^2 \sigma'((\mathbf{w}_r^{(L)})^\top \mathbf{x}_i^{(L-1)}) \sigma'((\mathbf{w}_r^{(L)})^\top \mathbf{x}_j^{(L-1)}) \cdot \xi_r^2
$$

Note that the tangent kernel $\mathbf{\Theta}_{\sigma,ij}^{(L)}(0)$ differs from $\mathbf{\Theta}_{\mu,ij}^{(L)}(0)$ with an additional term $\xi_r^2$. It is known the $\xi_r^2 \sim \chi_1$ independently with $\sigma'((\mathbf{w}_r^{(L)})^\top \mathbf{x}_i^{(L-1)} \geq 0)^\top \sigma'((\mathbf{w}_r^{(L)})^\top \mathbf{x}_j^{(L-1)} \geq 0)$.

Because $\mathbb{E}[\chi_1] = 1$, the expectation of $\mathbf{\Theta}_{\sigma,ij}^{(L)}(0)$ equals the expectation of $\mathbf{\Theta}_{\mu,ij}^{(L)}(0)$. Thus for all pairs of $i, j$, $\mathbf{\Theta}_{ij}^\sigma(0)$ is the average of $m$ i.i.d. random variables with the expectation

$$
\mathbb{E}\left[ \mathbf{\Theta}_\sigma^{(L)}(0) \right] = \mathbf{K}_\infty^{(L)}
$$

Now we calculate the concentration bound. It is known that $\xi_r^2$ is independent and sub-exponential. Then, by sub-exponential tail bound, we know that the following holds with probability at least $1 - \delta'$,

$$
\left| \mathbf{\Theta}_{\sigma,ij}^{(L)}(0) - \mathbf{K}_{\infty,ij}^{(L)} \right| \leq \sqrt{\frac{\log(8/\delta')}{2m}}
$$

This bound is of the same order to concentration bound for $\mathbf{\Theta}_{\mu,ij}^{(L)}(0)$. Thus we can take all the arguments for $\mathbf{\Theta}_{\mu,ij}^{(L)}(0)$ above to finalize the proof.

If $\sqrt{\frac{n^2 \log(8n^2/\delta)}{m}} \leq \frac{\lambda_{\min}(\mathbf{K}^{(L)})}{4}$, which implies $m = \Omega\left( \frac{n^2 \log(n^2/\delta)}{\lambda_{\min}^2(\mathbf{K}^{(L)})} \right)$, then with probability at least $1 - \delta$,

$$
\left\| \mathbf{\Theta}_\sigma^{(L)}(0) - \mathbf{K}_\infty^{(L)} \right\|_2 \leq \frac{\lambda_{\min}(\mathbf{K}^{(L)})}{4}
$$

**Phase 2, bounding $\left\| \mathbf{K}_\infty^{(L)} - \mathbf{K}^{(L)} \right\|_2$.** We show with probability $1 - \delta$ over the $\mathbf{W}^{(l)}$, for any $1 \leq l \leq L - 1, 1 \leq i, j \leq n$,

$$
\left\| \frac{1}{m} \sum_{r=1}^m (\mathbf{x}_i^{(l)})^\top \mathbf{x}_i^{(l)} - \mathbf{K}_{\infty,ij}^{(l)} \right\|_\infty \leq \mathcal{E} \sqrt{\frac{\log(Ln/\delta)}{m}}
$$

The error constant $\mathcal{E}$ depends on the choice of activation function, and satisfies

$$
\mathcal{E} \leq C \cdot 2^{O(L)}
$$

with $C$ being a positive constant. The $2^{O(L)}$ term comes form perturbation propagation through the neural network. The proof is by induction, and detailed proof can be found in the proof of Theorem E.1 in Du et al. (2019). Applying the union bound to the number of paths concludes the theorem, and the condition of $m$ follows:

$$m = \Omega\left(\frac{n^2 \log(Ln/\delta)2^{O(L)}}{\lambda_{\min}^2(\mathbf{K}^{(L)})}\right)$$

$\square$

**Remark A.1.** *The concentration bound is over two randomness, one is initialization of $\mathbf{W}_\mu$ and the other is Gaussian variable $\boldsymbol{\xi}$.*

## A.2 STEP 2. BOUNDING LEAST EIGENVALUE OF PNTK DURING TRAINING.

The next problem is that PNTKs are time-dependent matrices, thus varying during training. To account for this problem, we establish following lemmas stating that if the weight $\mathbf{W}^{(l)}(t)$ is close to $\mathbf{W}^{(l)}(0)$ during gradient descent training, then the corresponding PNTKs $\boldsymbol{\Theta}_\mu^{(L)}(t)$ and $\boldsymbol{\Theta}_\sigma^{(L)}(t)$ are close to their initialization $\boldsymbol{\Theta}_\mu^{(L)}(t)$, $\boldsymbol{\Theta}_\sigma^{(L)}(t)$ respectively.

Importantly, we introduce an auxiliary weight matrix $\widetilde{\mathbf{W}}^{(l)}(t) \equiv \mathbf{W}_\mu^{(l)}(t) + \mathbf{W}_\sigma^{(l)}(t) \odot \boldsymbol{\xi}(0)$, and an auxiliary weight vector $\widetilde{\mathbf{v}}(t) \equiv \mathbf{v}_\mu(t) + \mathbf{v}_\sigma(t) \odot \boldsymbol{\xi}^v(0)$, where $\boldsymbol{\xi}(0)$ and $\boldsymbol{\xi}^v(0)$ are the exact value of random variables at initialization. Then we demonstrate lemmas in step 2 as follows:

**Lemma A.4.** *If $\mathbf{W}_\mu(0)$ and $\mathbf{W}_\sigma(0)$ are initialized by the form described in Section 3.1, and suppose for every $l \in [L]$, $\|\mathbf{W}^{(l)}(0)\|_2 \le c_{w,0}\sqrt{m}$, $\|\mathbf{x}^{(l)}(0)\|_2 \le c_{x,0}$ and $\|\widetilde{\mathbf{W}}^{(l)}(t) - \mathbf{W}^{(l)}(0)\|_F \le \sqrt{m}R$ for some constant $c_{w,0}, c_{x,0} > 0$ and $R \le c_{w,0}$. If $\sigma(\cdot)$ is $H$-Lipschitz, then with probability at least $1 - \delta$, we have*

$$\|\mathbf{x}^{(l)}(t) - \mathbf{x}^{(l)}(0)\|_2 \le HRc_{x,0}g_{c_x}(l)(1 + \log(2/\delta))$$

*where $c_x = 2\sqrt{c_\sigma}Hc_{w,0}$.*

**Lemma A.5.** *If $\mathbf{W}_\mu(0)$ and $\mathbf{W}_\sigma(0)$ are initialized by the form described in Section 3.1, and uppose $\sigma(\cdot)$ is $H-$Lipschitz and $\beta-$smooth. Suppose for $l \in [L]$, $\|\mathbf{W}^{(l)}(0)\|_2 \le c_{w,0}\sqrt{m}$, $\|\mathbf{v}(0)\|_2 \le v_{2,0}\sqrt{m}$, $\|\mathbf{v}(0)\|_4 \le a_{4,0}m^{1/4}$, $\frac{1}{c_{x,0}} \le \|\mathbf{x}^{(l)}(0)\|_2 \le c_{x,0}$. If $\|\widetilde{\mathbf{W}}^{(l)}(t) - \mathbf{W}^{(l)}(0)\|_F, \|\widetilde{\mathbf{v}}(t) - \mathbf{v}(0)\|_2 \le \sqrt{m}R$ where $R \le cg_{c_x}(L)^{-1}\lambda_{\min}(\mathbf{K}^{(L)})n^{-1}(1 + \log(2/\delta))^{-2}$, $R \le cg_{c_x}(L)^{-1}\lambda_{\min}(\mathbf{K}^{(L)})n^{-1}(1 + \log(2/\delta))^{-3}$, and $R \le cg_{c_x}(L)^{-1}$ for some small constant $c$ and $c_x = 2\sqrt{c_\sigma}Hc_{w,0}$ then with probability at least $1 - \delta$, we have*

$$\left\|\boldsymbol{\Theta}_\mu^{(L)}(t) - \boldsymbol{\Theta}_\mu^{(L)}(0)\right\|_2 \le \frac{\lambda_{\min}(\mathbf{K}^{(L)})}{4}$$

$$\left\|\boldsymbol{\Theta}_\sigma^{(L)}(t) - \boldsymbol{\Theta}_\sigma^{(L)}(0)\right\|_2 \le \frac{\lambda_{\min}(\mathbf{K}^{(L)})}{4}$$

**Lemma A.6.** *If $\widehat{R}_S(Q, t') \le \exp(-\lambda_{\min}(\mathbf{K}^{(L)})t')\widehat{R}_S(Q, 0)$ holds for $0 \le t' \le t$, we have for any $0 \le s \le t$*

$$\left\|\widetilde{\mathbf{W}}^{(l)}(s) - \mathbf{W}^{(l)}(0)\right\|_F, \left\|\widetilde{\mathbf{v}}(s) - \mathbf{v}(0)\right\|_2 \le R'\sqrt{m}$$

*where $R' = \frac{16(1+\log(2/\delta))^2 c_{x,0}v_{2,0}(c_x)^L\sqrt{n}\|\mathbf{y}-f(\mathbf{X},Q(0))\|_2}{\lambda_0\sqrt{m}}$ for some small constant $c$ with $c_x = \max\{2\sqrt{c_\sigma}Lc_{w,0}, 1\}$.*

*Proof of Lemma A.4.* The proof sketch is by induction method.

For $l = 0$, where the target is input which is fixed, thus satisfying the hypothesis. Now suppose the induction hypothesis holds for $l' = 0, \ldots, l - 1$, we consider $l' = l$.

$$\left\|\mathbf{x}^{(l)}(t) - \mathbf{x}^{(l)}(0)\right\|_2 = \sqrt{\frac{1}{m}}\left\|\sigma(\mathbf{W}^{(l)}(t)\mathbf{x}^{(l-1)}(t)) - \sigma(\mathbf{W}^{(l)}(0)\mathbf{x}^{(l-1)}(0))\right\|_2$$

$$\leq \sqrt{\frac{1}{m}}\left\|\sigma(\mathbf{W}^{(l)}(t)\mathbf{x}^{(l-1)}(t)) - \sigma(\mathbf{W}^{(l)}(t)\mathbf{x}^{(l-1)}(0))\right\|_2$$

$$+ \sqrt{\frac{1}{m}}\left\|\sigma(\mathbf{W}^{(l)}(t)\mathbf{x}^{(l-1)}(0)) - \sigma(\mathbf{W}^{(l)}(0)\mathbf{x}^{(l-1)}(0))\right\|_2$$

$$\leq \sqrt{\frac{1}{m}}H\left(\left\|\mathbf{W}^{(l)}(0)\right\|_2 + \left\|\mathbf{W}^{(l)}(t) - \widetilde{\mathbf{W}}^{(l)}(t)\right\|_2 + \left\|\widetilde{\mathbf{W}}^{(l)}(t) - \mathbf{W}^{(l)}(0)\right\|_F\right)$$

$$\cdot \left\|\mathbf{x}^{(l-1)}(t) - \mathbf{x}^{(l-1)}(0)\right\|_2$$

$$+ \sqrt{\frac{1}{m}}H\left(\left\|\mathbf{W}^{(l)}(t) - \widetilde{\mathbf{W}}^{(l)}(t)\right\|_2 + \left\|\widetilde{\mathbf{W}}^{(l)}(t) - \mathbf{W}^{(l)}(0)\right\|_F\right)\left\|\mathbf{x}^{h-1}(0)\right\|_2$$

$$\leq \sqrt{\frac{1}{m}}H\left(c_{w,0}\sqrt{m} + R\sqrt{m}(1 + \log(2/\delta))\right)HRc_{x,0}g_{c_x}(l-1)$$

$$+ \sqrt{\frac{1}{m}}H\sqrt{m}R(1 + \log(2/\delta)c_{x,0}$$

$$\leq HRc_{x,0}\left(c_x g_{c_x}(l-1) + 1\right)(1 + \log(2/\delta))$$

$$\leq HRc_{x,0}g_{c_x}(l)(1 + \log(2/\delta))$$

$$\square$$

*Proof of Lemma A.5.* For simplicity we define $z_{i,r}(t) = \mathbf{w}_r^{(L)}(t)^\top \mathbf{x}_i^{(L-1)}(t)$.

Now we bound the distance between $\boldsymbol{\Theta}_{\mu,ij}^{(L)}(t)$ and $\boldsymbol{\Theta}_{\mu,ij}^{(L)}(0)$ through the following inequality:

$$\left|\boldsymbol{\Theta}_{\mu,ij}^{(L)}(t) - \boldsymbol{\Theta}_{\mu,ij}^{(L)}(0)\right|$$

$$= \left|\mathbf{x}_i^{(L-1)}(t)^\top\mathbf{x}_j^{(L-1)}(t)\frac{1}{m}\sum_{r=1}^m v_r(t)^2\sigma'\left(z_{i,r}(t)\right)\sigma'\left(z_{j,r}(t)\right)\right.$$

$$\left. - \mathbf{x}_i^{(L-1)}(0)^\top\mathbf{x}_j^{(L-1)}(0)\frac{1}{m}\sum_{r=1}^m v_r(0)^2\sigma'\left(z_{i,r}(0)\right)\sigma'\left(z_{j,r}(0)\right)\right|$$

$$\leq \left|\mathbf{x}_i^{(L-1)}(t)^\top\mathbf{x}_j^{(L-1)}(t) - \mathbf{x}_i^{(L-1)}(0)^\top\mathbf{x}_j^{(L-1)}(0)\right|\frac{1}{m}\sum_{r=1}^m v_r(0)^2\left|\sigma'\left(z_{i,r}(t)\right)\sigma'\left(z_{j,r}(t)\right)\right|$$

$$+ \left|\mathbf{x}_i^{(L-1)}(0)^\top\mathbf{x}_j^{(L-1)}(0)\right|\frac{1}{m}\left|\sum_{r=1}^m v_r(0)^2\left(\sigma'\left(z_{i,r}(t)\right)\sigma'\left(z_{j,r}(t)\right) - \sigma'\left(z_{i,r}(0)\right)\sigma'\left(z_{j,r}(0)\right)\right)\right|$$

$$+ \left|\mathbf{x}_i^{(L-1)}(t)^\top\mathbf{x}_j^{(L-1)}(t)\right|\frac{1}{m}\left|\sum_{r=1}^m \left(v_r(t)^2 - v_r(0)^2\right)\sigma'\left(z_{i,r}(t)\right)\sigma'\left(z_{j,r}(t)\right)\right|$$

$$\leq H^2v_{2,0}^2\left|\mathbf{x}_i^{(L-1)}(t)^\top\mathbf{x}_j^{(L-1)}(t) - \mathbf{x}_i^{(L-1)}(0)^\top\mathbf{x}_j^{(L-1)}(0)\right|$$

$$+ c_{x,0}^2\frac{1}{m}\left|\sum_{r=1}^m v_r(0)^2\left(\sigma'\left(z_{i,r}(t)\right)\sigma'\left(z_{j,r}(t)\right) - \sigma'\left(z_{i,r}(0)\right)\sigma'\left(z_{j,r}(0)\right)\right)\right|$$

$$+ 4H^2c_{x,0}^2\frac{1}{m}\sum_{r=1}^m\left|v_r(t)^2 - v_r(0)^2\right|$$

$$\equiv I_1^{i,j} + I_2^{i,j} + I_3^{i,j}.$$

For $I_1^{i,j}$, by Lemma A.4, we have

$$
\begin{aligned}
I_1^{i,j} =& H^2 v_{2,0}^2 \left| \mathbf{x}_i^{(L-1)}(t)^\top \mathbf{x}_j^{(L-1)}(t) - \mathbf{x}_i^{(L-1)}(0)^\top \mathbf{x}_j^{(L-1)}(0) \right| \\
\leq& H^2 v_{2,0}^2 \left| (\mathbf{x}_i^{(L-1)}(t) - \mathbf{x}_i^{(L-1)}(0))^\top \mathbf{x}_j^{(L-1)}(t) \right| + H^2 v_{2,0}^2 \left| \mathbf{x}_i^{(L-1)}(0)^\top (\mathbf{x}_j^{(L-1)}(t) - \mathbf{x}_j^{(L-1)}(0)) \right| \\
\leq& v_{2,0}^2 H^3 c_{x,0} g_{c_x}(L) R(1 + \log(2/\delta)) \cdot (c_{x,0} + H c_{x,0} g_{c_x}(L) R(1 + \log(2/\delta))) \\
& + v_{2,0}^2 H^3 c_{x,0} g_{c_x}(L) R c_{x,0}(1 + \log(2/\delta)) \\
\leq& 3 v_{2,0}^2 c_{x,0}^2 H^3 g_{c_x}(L) R(1 + \log(2/\delta))^2
\end{aligned}
$$

For $I_2^{i,j}$, we have

$$
\begin{aligned}
I_2^{i,j} =& c_{x,0}^2 \frac{1}{m} \left| \sum_{r=1}^m v_r(0)^2 \sigma'\left(z_{i,r}(t)\right) \sigma'\left(z_{j,r}(t)\right) - v_r(0)^2 \sigma'\left(z_{i,r}(0)\right) \sigma'\left(z_{j,r}(0)\right) \right| \\
\leq& c_{x,0}^2 \frac{1}{m} \sum_{r=1}^m v_r(0)^2 \left| \left(\sigma'\left(z_{i,r}(t)\right) - \sigma'\left(z_{i,r}(0)\right)\right) \sigma'\left(z_{j,r}(t)\right) \right| \\
& + v_r(0)^2 \left| \left(\sigma'\left(z_{j,r}(t)\right) - \sigma'\left(z_{j,r}(0)\right)\right) \sigma'\left(z_{i,r}(0)\right) \right| \\
\leq& \frac{\beta H c_{x,0}^2}{m} \left( \sum_{r=1}^m v_r(0)^2 \left| z_{i,r}(t) - z_{i,r}(0) \right| + v_r(0)^2 \left| z_{j,r}(t) - z_{j,r}(0) \right| \right) \\
\leq& \frac{\beta H v_{4,0}^2 c_{x,0}^2}{\sqrt{m}} \left( \sqrt{\sum_{r=1}^m \left| z_{i,r}(t) - z_{i,r}(0) \right|^2} + \sqrt{\sum_{r=1}^m \left| z_{j,r}(t) - z_{j,r}(0) \right|^2} \right).
\end{aligned}
$$

Using the same proof for Lemma A.4, it is easy to see that

$$
\sum_{r=1}^m \left| z_{i,r}(t) - z_{i,r}(0) \right|^2 \leq c_{x,0}^2 g_{c_x}(L)^2 m R^2 (1 + \log(2/\delta))^2.
$$

Thus

$$
I_2^{i,j} \leq 2\beta v_{4,0}^2 c_{x,0}^3 L g_{c_x}(L) R(1 + \log(2/\delta)).
$$

For $I_3^{i,j}$,

$$
\begin{aligned}
I_3^{i,j} =& 4 H^2 c_{x,0}^2 \frac{1}{m} \sum_{r=1}^m \left| v_r(t)^2 - v_r(0)^2 \right| \\
\leq& 4 H^2 c_{x,0}^2 \frac{1}{m} \sum_{r=1}^m \left| v_r(t) - v_r(0) \right| \left| v_r(t) \right| + \left| v_r(t) - v_r(0) \right| \left| v_r(0) \right| \\
\leq& 12 H^2 c_{x,0}^2 v_{2,0} R(1 + \log(2/\delta)).
\end{aligned}
$$

Therefore we can bound the perturbation

$$
\begin{aligned}
\left\| \boldsymbol{\Theta}_\mu^{(L)}(t) - \boldsymbol{\Theta}_\mu^{(L)}(0) \right\|_F =& \sqrt{\sum_{i,j=1}^n \left| \boldsymbol{\Theta}_{\mu,ij}^{(L)}(t) - \boldsymbol{\Theta}_{\mu,ij}^{(L)}(0) \right|^2} \\
\leq& \big[ \left(2\beta c_{x,0} v_{4,0}^2 + 3 H^2\right) H c_{x,0}^2 v_{2,0}^2 g_{c_x}(L)(1 + \log(2/\delta))^2 \\
& + 12 H^2 c_{x,0}^2 v_{2,0}(1 + \log(2/\delta)) \big] n R
\end{aligned}
$$

Recall the bound on $R$, which is $R \leq cg_{c_x}(L)^{-1}\lambda_{\min}(\mathbf{K}^{(L)})n^{-1}(1 + \log(2/\delta))^{-2}$, we have the desired result for $\mathbf{\Theta}_\mu^{(L)}$:

$$\left\| \mathbf{\Theta}_\mu^{(L)}(t) - \mathbf{\Theta}_\mu^{(L)}(0) \right\|_2 \leq \frac{\lambda_{\min}(\mathbf{K}^{(L)})}{4}$$

Then we bound the distance between $\mathbf{\Theta}_{\sigma,ij}^{(L)}(t)$ and $\mathbf{\Theta}_{\sigma,ij}^{(L)}(0)$ through the following inequality:

$$\left| \mathbf{\Theta}_{\sigma,ij}^{(L)}(t) - \mathbf{\Theta}_{\sigma,ij}^{(L)}(0) \right|$$

$$= \left| \mathbf{x}_i^{(L-1)}(t)^\top \mathbf{x}_j^{(L-1)}(t) \frac{1}{m} \sum_{r=1}^m v_r(t)^2 \sigma'\left(z_{i,r}(t)\right) \sigma'\left(z_{j,r}(t)\right) \cdot \xi_r^2(t) \right.$$

$$\left. - \mathbf{x}_i^{(L-1)}(0)^\top \mathbf{x}_j^{(L-1)}(0) \frac{1}{m} \sum_{r=1}^m v_r(0)^2 \sigma'\left(z_{i,r}(0)\right) \sigma'\left(z_{j,r}(0) \cdot \xi_r^2(0)\right) \right|$$

$$\leq \left| \mathbf{x}_i^{(L-1)}(t)^\top \mathbf{x}_j^{(L-1)}(t) - \mathbf{x}_i^{(L-1)}(0)^\top \mathbf{x}_j^{(L-1)}(0) \right| \frac{1}{m} \sum_{r=1}^m v_r(0)^2 \left| \sigma'\left(z_{i,r}(t)\right) \sigma'\left(z_{j,r}(t)\right) \right| \cdot \xi_r^2$$

$$+ \left| \mathbf{x}_i^{(L-1)}(0)^\top \mathbf{x}_j^{(L-1)}(0) \right| \frac{1}{m} \left| \sum_{r=1}^m v_r(0)^2 \left( \sigma'\left(z_{i,r}(t)\right) \sigma'\left(z_{j,r}(t)\right) - \sigma'\left(z_{i,r}(0)\right) \sigma'\left(z_{j,r}(0)\right) \right) \right|$$

$$+ \left| \mathbf{x}_i^{(L-1)}(t)^\top \mathbf{x}_j^{(L-1)}(t) \right| \frac{1}{m} \left| \sum_{r=1}^m \left( v_r(t)^2 - v_r(0)^2 \right) \sigma'\left(z_{i,r}(t)\right) \sigma'\left(z_{j,r}(t)\right) \cdot \xi_r^2(t) \right|$$

$$+ \left| \mathbf{x}_i^{(L-1)}(0)^\top \mathbf{x}_j^{(L-1)}(0) \right| \frac{1}{m} \left| \sum_{r=1}^m v_r(0)^2 \left( \sigma'\left(z_{i,r}(0)\right) \sigma'\left(z_{j,r}(0)\right) \right) \left( \xi_r^2(t) - \xi_r^2(0) \right) \right|$$

$$\leq H^2 v_{2,0}^2 (1 + \log(2/\delta)) \left| \mathbf{x}_i^{(L-1)}(t)^\top \mathbf{x}_j^{(L-1)}(t) - \mathbf{x}_i^{(L-1)}(0)^\top \mathbf{x}_j^{(L-1)}(0) \right| (1 + \log(2/\delta))$$

$$+ c_{x,0}^2 (1 + \log(2/\delta)) \frac{1}{m} \left| \sum_{r=1}^m v_r(0)^2 \left( \sigma'\left(z_{i,r}(t)\right) \sigma'\left(z_{j,r}(t)\right) - \sigma'\left(z_{i,r}(0)\right) \sigma'\left(z_{j,r}(0)\right) \right) \right|$$

$$+ 4H^2 c_{x,0}^2 (1 + \log(2/\delta)) \frac{1}{m} \sum_{r=1}^m \left| v_r(t)^2 - v_r(0)^2 \right|$$

$$+ 4H^2 c_{x,0}^2 v_{2,0}^2 \frac{1}{m} \sum_{r=1}^m \left| \xi_r(t)^2 - \xi_r(0)^2 \right|$$

$$= (1 + \log(2/\delta))(I_1^{i,j} + I_2^{i,j} + I_3^{i,j}) + I_4^{i,j}.$$

For $I_4^{i,j}$, by the tail bound fora chi-square variable, we have

$$I_4^{i,j} \leq 4H^2 c_{x,0}^2 v_{2,0}^2 \left( 1 + \sqrt{\frac{\log(2/\delta)}{m}} \right)$$

Therefore we can bound the perturbation

$$\left\| \mathbf{\Theta}_\sigma^{(L)}(t) - \mathbf{\Theta}_\sigma^{(L)}(0) \right\|_F = \sqrt{\sum_{i,j=1}^n \left| \mathbf{\Theta}_{\sigma,ij}^{(L)}(t) - \mathbf{\Theta}_{\sigma,ij}^{(L)}(0) \right|^2}$$

$$\leq \Big[ \left( 2\beta c_{x,0} v_{4,0}^2 + 3H^2 \right) H c_{x,0}^2 v_{2,0}^2 g_{c_x}(L)(1 + \log(2/\delta))^3$$

$$+ 12H^2 c_{x,0}^2 v_{2,0}(1 + \log(2/\delta))^2 + 4H^2 c_{x,0}^2 v_{2,0}^2 (1 + \sqrt{\frac{\log(2/\delta)}{m}}) \Big] nR$$

Recall the bound on $R$, which is $R \leq cg_{c_x}(L)^{-1}\lambda_{\min}(\mathbf{K}^{(L)})n^{-1}(1 + \log(2/\delta))^{-3}$, we have the desired result for $\boldsymbol{\Theta}_\sigma^{(L)}$:

$$\left\|\boldsymbol{\Theta}_\sigma^{(L)}(t) - \boldsymbol{\Theta}_\sigma^{(L)}(0)\right\|_2 \leq \frac{\lambda_{\min}(\mathbf{K}^{(L)})}{4}$$

$\square$

*Proof of Lemma A.6.* We first consider the derivative of $\mathbf{W}_\mu^{(l)}$ and have:

$$\left\|\frac{d}{ds}\mathbf{W}_\mu^{(l)}(s)\right\|_F$$

$$=\eta\left\|\left(\frac{1}{m}\right)^{\frac{L-l+1}{2}}\sum_{i=1}^n (y_i - f(\mathbf{x}_i; s))\mathbf{x}_i^{(l-1)}(s)\left(\mathbf{v}(s)^\top\left(\prod_{k=l+1}^L \mathbf{J}_i^{(k)}(s)\mathbf{W}^{(k)}(s)\right)\mathbf{J}_i^{(l)}(s)\right)\right\|_F$$

$$\leq\eta\left(\frac{1}{m}\right)^{\frac{L-l+1}{2}}\|\mathbf{v}(s)\|_2\sum_{i=1}^n |y_i - f(\mathbf{x}_i; s)|\|\mathbf{x}_i^{(l-1)}(s)\|_2\prod_{k=l+1}^L\|\mathbf{W}^{(k)}(s)\|_2\prod_{k=l}^L\|\mathbf{J}^{(k)}(s)\|_2,$$

$$\left\|\frac{d}{ds}\mathbf{v}_\mu(s)\right\|_2 = \eta\left\|\sum_{i=1}^n (y_i - f(\mathbf{x}_i; s))\mathbf{x}_i^{(L)}(s)\right\|_2.$$

where

$$\mathbf{J}^{(l')} \equiv \mathrm{diag}\left(\sigma'\left((\mathbf{w}_1^{(l')})^\top\mathbf{x}^{(l'-1)}\right), \ldots, \sigma'\left((\mathbf{w}_m^{(l')})^\top\mathbf{x}^{(l'-1)}\right)\right) \in \mathbb{R}^{m\times m}$$

are the derivative matrices induced by the activation function.

To bound $\left\|\mathbf{x}_i^{(l-1)}(s)\right\|_2$, we can just apply Lemma A.4 and get

$$\left\|\mathbf{x}_i^{(l-1)}(s)\right\|_2 \leq Hc_{x,0}g_{c_x}(h)R(1 + \log(2/\delta)) + c_{x,0} \leq 2c_{x,0}(1 + \log(2/\delta)).$$

To bound $\left\|\mathbf{W}^{(k)}(s)\right\|_2$, we use our assumption

$$\prod_{k=l+1}^L \|\mathbf{W}^{(k)}(s)\|_2 \leq \prod_{k=l+1}^L\left(\|\mathbf{W}^{(k)}(0)\|_2 + \|\mathbf{W}^{(k)}(s) - \mathbf{W}^{(k)}(0)\|_2\right)$$

$$\leq \prod_{k=l+1}^L (c_{w,0}\sqrt{m} + R'\sqrt{m})(1 + \log(2/\delta))$$

$$= (c_{w,0} + R')^{L-l} m^{\frac{L-l}{2}}(1 + \log(2/\delta))$$

$$\leq (2c_{w,0})^{L-l} m^{\frac{L-l}{2}}(1 + \log(2/\delta)).$$

Note that $\left\|\mathbf{J}^{(k)}(s)\right\|_2 \leq H$. Plugging in these two bounds back, we obtain

$$\left\|\frac{d}{ds}\mathbf{W}_\mu^{(l)}(s)\right\|_F \leq 4\eta c_{x,0}v_{2,0}c_x^L(1 + \log(2/\delta))^2\sum_{i=1}^n |y_i - f(\mathbf{x}_i; s)|$$

$$\leq 4\eta c_{x,0}v_{2,0}c_x^L(1 + \log(2/\delta))^2\sqrt{n}\|\mathbf{y} - f(\mathbf{X}; s)\|_2$$

$$\leq (1 + \log(2/\delta))^2 e^{-\lambda_0 s}\frac{1}{4}\eta\lambda_0 R'\sqrt{m}.$$

Similarly, we have

$$\left\|\frac{d}{dt}\mathbf{v}_\mu(s)\right\|_2 \leq 2\eta c_{x,0}\sum_{i=1}^n |y_i - f(\mathbf{x}_i; s)|$$

$$\leq e^{-\lambda_0 s}\frac{1}{4}\eta\lambda_0 R'\sqrt{m}.$$

Next, we consider the derivative of $\mathbf{W}_\sigma^{(l)}$ and have:

$$\left\| \frac{d}{ds} \mathbf{W}_\sigma^{(l)}(s) \right\|_F = \eta \left\| \left( \frac{1}{m} \right)^{\frac{L-l+1}{2}} \mathbb{E}_{Q(s)} \left[ \sum_{i=1}^{n} (y_i - f(\mathbf{x}_i, Q(s))) \mathbf{x}_i^{(l-1)}(s) \odot \boldsymbol{\xi}^{(l)} \right] \right.$$

$$\left. \left( \mathbf{v}(s)^\top \left( \prod_{k=l+1}^{L} \mathbf{J}_i^{(k)}(s) \mathbf{W}^{(k)}(s) \right) \mathbf{J}_i^{(l)}(s) \right) \right\|_F$$

$$= 0$$

where we have used we use the definition of loss $\widehat{R}_S(Q) = \mathbb{E}_{f \sim Q} \widehat{R}_S(f)$ and interchanged integration and differentiation. Similarly, for $\mathbf{v}_\sigma$ we have,

$$\left\| \frac{d}{ds} \mathbf{v}_\sigma(s) \right\|_2 = \eta \left\| \mathbb{E}_{Q(s)} \left[ \sum_{i=1}^{n} (y_i - f(\mathbf{x}_i; s)) \mathbf{x}_i^{(L)}(s) \odot \boldsymbol{\xi}^{(v)} \right] \right\|_2 = 0$$

Integrating the derivative of weights, we obtain

$$\left\| \widetilde{\mathbf{W}}^{(l)}(s) - \mathbf{W}^{(l)}(0) \right\|_F \leq \left\| \mathbf{W}_\mu^{(l)}(s) - \mathbf{W}_\mu^{(l)}(0) \right\|_F + \left\| \left( \mathbf{W}_\sigma^{(l)}(s) - \mathbf{W}_\sigma^{(l)}(0) \right) \cdot \boldsymbol{\xi}^{(l)}(0) \right\|_F$$

$$\leq \int_{s'=0}^{s} \left\| \frac{d}{ds'} \mathbf{W}_\mu^{(l)}(s') \right\|_F \leq R'\sqrt{m}$$

$$\left\| \widetilde{\mathbf{v}}(s) - \mathbf{v}(0) \right\|_2 \leq \int_{s'=0}^{s} \left\| \frac{d}{ds'} \mathbf{v}_\mu(s') \right\|_2 \leq R'\sqrt{m}$$

$\square$

## A.3 SETP 3. TOWARDS LINEAR CONVERGENCE RATE OF EMPIRICAL LOSS

Now we process to analyze the convergence rate of empirical error. Combined with fact that least eigenvalue of PNTKs and change of weights are bounded during training, the behavior of the loss is traceable. To finalize the proof for Theorem 4.2, we show:

**Lemma A.7.** *If $R' < R$, we have $\widehat{R}_S(Q(t)) \leq \exp(-\lambda_0 t)\widehat{R}_S(Q(0))$.*

*Proof of Lemma A.7.* According to the gradient flow of output function, we have

$$\frac{df(\mathbf{x}_i, t)}{dt} = \sum_{l=1}^{L} \left( \left\langle \frac{\partial f(\mathbf{x}_i; t)}{\partial \mathbf{W}_\mu^{(l)}}, \frac{d\mathbf{W}_\mu^{(l)}(t)}{dt} \right\rangle + \left\langle \frac{\partial f(\mathbf{x}; t)}{\partial \mathbf{W}_\sigma^{(l)}}, \frac{d\mathbf{W}_\sigma^{(l)}(t)}{dt} \right\rangle \right)$$

$$+ \left\langle \frac{\partial f(\mathbf{x}_i; t)}{\partial \mathbf{v}_\mu}, \frac{d\mathbf{v}_\mu(t)}{dt} \right\rangle + \left\langle \frac{\partial f(\mathbf{x}; t)}{\partial \mathbf{v}_\sigma}, \frac{d\mathbf{v}_\sigma(t)}{dt} \right\rangle$$

$$= \sum_{j=1}^{n} (\mathbf{y}_i - f(\mathbf{x}_j)) \left[ \sum_{l=1}^{L} \left( \left\langle \frac{\partial f(\mathbf{x}_i)}{\partial \mathbf{W}_\mu^{(l)}}, \frac{f(\mathbf{x}_j)}{\partial \mathbf{W}_\mu^{(l)}} \right\rangle + \left\langle \frac{\partial f(\mathbf{x}_i)}{\partial \mathbf{W}_\sigma^{(l)}}, \frac{\partial f(\mathbf{x}_j)}{\partial \mathbf{W}_\sigma^{(l)}} \right\rangle \right) \right.$$

$$+ \left\langle \frac{\partial f(\mathbf{x}_i)}{\partial \mathbf{v}_\mu}, \frac{f(\mathbf{x}_j)}{\partial \mathbf{v}_\mu} \right\rangle + \left\langle \frac{\partial f(\mathbf{x}_i)}{\partial \mathbf{v}_\sigma}, \frac{\partial f(\mathbf{x}_j)}{\partial \mathbf{v}_\sigma} \right\rangle \right]$$

$$\geq \sum_{j=1}^{n} (\mathbf{y}_j - f(\mathbf{x}_j; t))(\mathbf{\Theta}_{\mu,ij}^{(L)} + \mathbf{\Theta}_{\sigma,ij}^{(L)})$$

Then the dynamics of loss can be calculated,

$$\frac{d}{dt}\widehat{R}_S(Q(t)) = \frac{1}{2}\frac{d}{dt}\left\|\mathbb{E}_{f\sim Q(t)}f(\mathbf{X};t) - \mathbf{y}\right\|_2^2$$

$$\leq -\Big(\mathbf{y} - f(\mathbf{X},Q(t))\Big)^\top\Big(\boldsymbol{\Theta}_\mu^{(L)}(t) + \boldsymbol{\Theta}_\sigma^{(L)}(t)\Big)\Big(\mathbf{y} - f(\mathbf{X},Q(t))\Big)$$

$$\leq -\lambda_0\Big\|\mathbf{y} - \mathbb{E}_{f\sim Q(t)}f(\mathbf{X};t)\Big\|_2^2$$

where we have used the condition $R' < R$. Therefore, we have the desired result:

$$\widehat{R}_S(Q,t) \leq \exp(-\lambda_0 t)\widehat{R}_S(Q,0)$$

Finally, we provide a bound for $\widehat{R}_S(Q(0))$:

$$\left\|\mathbf{y} - f(\mathbf{X},Q(0))\right\|_2^2 = \sum_{i=1}^n\Big(y_i^2 + y_i f(\mathbf{x}_i,Q(0)) + f(\mathbf{x}_i,Q(0))^2]\Big)$$

$$= \sum_{i=1}^n(1 + O(1))$$

$$= O(n)$$

Recall that in Lemma A.5 and Lemma A.6:

$$R \leq cg_{c_x}(L)^{-1}\lambda_{\min}(\mathbf{K}^{(L)})n^{-1}(1 + \log(2/\delta))^{-3}$$

$$R' = \frac{16(1 + \log(2/\delta))^2 c_{x,0} v_{2,0}(c_x)^L\sqrt{n}\|\mathbf{y} - f(\mathbf{X},Q(0))\|_2}{\lambda_0\sqrt{m}}$$

Thus $R' < R$ yields

$$m = \Omega\left(\frac{n^5\log(2/\delta)^{10}}{\lambda_0^4}\right)$$

$\square$

## B    PROOF OF THEOREM 4.3

**Theorem B.1** (Restatement of Theorem 4.3). *Consider gradient descent on objective function (6). Suppose $m \geq \mathrm{poly}(n, 1/\lambda_0, 1/\delta, 1/\mathcal{E})$. Then, with a probability of at least $1 - \delta$ over the random initialization, we have*

$$f(\mathbf{x},Q(t))|_{t=\infty} = \boldsymbol{\Theta}_\mu^\infty(\mathbf{x},\mathbf{X})\big(\boldsymbol{\Theta}_\mu^\infty(\mathbf{X},\mathbf{X}) + \lambda/c_\sigma^2\mathbf{I}\big)^{-1}\mathbf{y} \pm \mathcal{E}$$

*where $f(\mathbf{x},Q(t)) = \mathbb{E}_{f\sim Q(t)}f(\mathbf{x};t)$ aligns with the definition of the empirical loss function.*

*Proof of Theorem B.1.* To proceed the proof, we first establish the result of kernel ridge regression in the infinite-width limit, and then bound the perturbation on the predict. According the linearization rules for infinitely-wide networks (Lee et al., 2019), the output function can be expressed as,

$$f_{\mathrm{ntk}}(\mathbf{x},t) = \phi_\mu(\mathbf{x})^\top(\boldsymbol{\theta}_\mu(t) - \boldsymbol{\theta}_\mu(0)) + \phi_\sigma(\mathbf{x})^\top(\boldsymbol{\theta}_\sigma(t) - \boldsymbol{\theta}_\sigma(0)),$$

where $\phi_\mu(\mathbf{x}) = \nabla_{\boldsymbol{\theta}_\mu}f(\mathbf{x},Q(0))$, and $\phi_\sigma(\mathbf{x}) = \nabla_{\boldsymbol{\theta}_\sigma}f(\mathbf{x},Q(0))$. Recall that $\boldsymbol{\theta}_\sigma$ does not change during training, as calculated in the proof of Lemma A.6, then the KL divergence reduces to

$$\mathrm{KL} = \frac{1}{2}\frac{(\theta_\mu(t) - \theta_\mu(0))^2}{c_\sigma^2}$$

Then the gradient flow equation for $\boldsymbol{\theta})\mu$ becomes,

$$
\begin{aligned}
\frac{d\boldsymbol{\theta}_\mu(t)}{dt} &= \frac{\partial L(Q(t))}{\partial \boldsymbol{\theta}_\mu} \\
&= \big(f_{\mathrm{ntk}}(\mathbf{X}, Q(t)) - \mathbf{y}\big)\phi_\mu(\mathbf{X}) + \lambda/c_\sigma^2\big(\boldsymbol{\theta}_\mu(t) - \boldsymbol{\theta}_\mu(0)\big) \\
&= \boldsymbol{\Theta}_\mu^\infty(\mathbf{X}, \mathbf{X})\big(\boldsymbol{\theta}_\mu(t) - \boldsymbol{\theta}_\mu(0)\big) - \phi_\mu(\mathbf{X})^\top \mathbf{y} + \lambda/c_\sigma^2\big(\boldsymbol{\theta}_\mu(t) - \boldsymbol{\theta}_\mu(0)\big)
\end{aligned}
$$

which is an ordinary differential equation regarding $\boldsymbol{\theta}_\mu(t)$, and the solution is,

$$
\boldsymbol{\theta}_\mu(t) = \phi_\mu(\mathbf{X})^\top\big(\boldsymbol{\Theta}_\mu^\infty(\mathbf{X}, \mathbf{X}) + \lambda/c_\sigma^2\mathbf{I}\big)^{-1}\mathbf{y}\big(\mathbf{I} - e^{-(\boldsymbol{\Theta}_\mu^\infty(\mathbf{X},\mathbf{X})+\lambda/c_\sigma^2\mathbf{I})t}\big)
$$

Plug this result into the linearization of expected output function, we have,

$$
f_{\mathrm{ntk}}(\mathbf{x}, t) = \boldsymbol{\Theta}_\mu^\infty(\mathbf{x}, \mathbf{X})(\boldsymbol{\Theta}_\mu^\infty(\mathbf{X}, \mathbf{X}) + \lambda/c_\sigma^2\mathbf{I})^{-1}\mathbf{y}(\mathbf{I} - e^{-(\boldsymbol{\Theta}_\mu^\infty(\mathbf{X},\mathbf{X})+\lambda/c_\sigma^2\mathbf{I})t})
$$

Then we take the time to be infinity and have

$$
f_{\mathrm{ntk}}(\mathbf{x})|_{t=\infty} = \boldsymbol{\Theta}_\mu^\infty(\mathbf{x}, \mathbf{X})(\boldsymbol{\Theta}_\mu^\infty(\mathbf{X}, \mathbf{X}) + \lambda/c_\sigma^2\mathbf{I})^{-1}\mathbf{y}.
$$

The next step is to show that

$$
\big|f(\mathbf{x}, Q(t)) - f_{\mathrm{ntk}}(\mathbf{x})\big| \le O(\mathcal{E}).
$$

where $\mathcal{E} = \mathcal{E}_{\mathrm{init}} + \frac{\sqrt{n}}{\lambda_0^2}\log(\frac{n}{\mathcal{E}_\Theta\lambda_0})\mathcal{E}_\Theta$ with $|f(\mathbf{x}, Q(0)| \le \mathcal{E}_{\mathrm{init}}$ and $\|\lim_{m\to\infty}\boldsymbol{\Theta}_\mu - \boldsymbol{\Theta}_\mu(t)\|_2 \le \mathcal{E}_\Theta$.

The proof relies a careful analysis on the trajectories induced by gradient flows for optimizing the neural network and the NTK predictor. The detailed proof can be found in the proof of Theorem 3.2 in Arora et al. (2019b), and we can replace kernel ridge regression here by kernel regression.

$\square$

## C  PROOFS OF SECTION 4.3

### C.1  PROOF OF THEOREM 4.4

**Theorem C.1** (Restatement of Theorem 4.4). *Suppose data $S = \{(\mathbf{x}_i, y_i)\}_{i=1}^n$ are i.i.d. samples from a non-degenerate distribution $\mathcal{D}$, and $m \ge \mathrm{poly}(n, \lambda_0^{-1}, \delta^{-1})$. Consider any loss function $\ell : \mathbb{R} \times \mathbb{R} \to [0, 1]$ that is 1-Lipschitz in the first argument such that $\ell(y, y) = 0$. Then with a probability of at least $1 - \delta$ over the random initialization and the training samples, the probabilistic neural network (PNN) trained by gradient descent for $T \ge \Omega(\frac{1}{\eta\lambda_0}\log\frac{n}{\delta})$ iterations has population risk $R_\mathcal{D}(Q)$ that is bounded as follows:*

$$
R_\mathcal{D}(Q) \le \frac{\mathbf{y}^\top(\boldsymbol{\Theta}_\mu^\infty(\mathbf{X}, \mathbf{X}) + \lambda/c_\sigma^2\mathbf{I})^{-1}\mathbf{y}}{nc_\sigma^2} + \frac{\lambda}{c_\sigma^2}\sqrt{\frac{\mathbf{y}^\top(\boldsymbol{\Theta}_\mu^\infty(\mathbf{X}, \mathbf{X}) + \lambda/c_\sigma^2\mathbf{I})^{-2}\mathbf{y}}{n}} + O\left(\frac{\log\frac{2\sqrt{n}}{\delta}}{n}\right).
$$

*Proof of Theorem C.1.* The generalization bound consists two terms, one is the empirical error, and another is KL divergence.

(1) We first bound the empirical error $\sqrt{\sum_{i=1}^n (f_{\mathrm{ntk}}(\mathbf{x}_i, Q(t = \infty)) - y_i)^2}$ with following inequality,

$$
\begin{aligned}
\sqrt{\sum_{i=1}^n (f_{\mathrm{ntk}}(\mathbf{x}_i, Q(t = \infty)) - y_i)^2} &= \left\|\boldsymbol{\Theta}_\mu^\infty(\mathbf{X}, \mathbf{X})(\boldsymbol{\Theta}_\mu^\infty(\mathbf{X}, \mathbf{X}) + \lambda/c_\sigma^2\mathbf{I})^{-1}\mathbf{y} - \mathbf{y}\right\|_2 \\
&= \left\|\lambda/c_\sigma^2\big(\boldsymbol{\Theta}_\mu^\infty(\mathbf{X}, \mathbf{X}) + \lambda/c_\sigma^2\mathbf{I}\big)^{-1}\mathbf{y}\right\|_2 \\
&= \lambda/c_\sigma^2\sqrt{\mathbf{y}^\top\big(\boldsymbol{\Theta}_\mu^\infty(\mathbf{X}, \mathbf{X}) + \lambda/\sigma_0^2\mathbf{I}\big)^{-2}\mathbf{y}}
\end{aligned}
$$

Then we can further bound the error term as follows:

$$\frac{1}{n}\sum_{i=1}^{n}\ell\big(f_{\mathrm{ntk}}(\mathbf{x}_i),y_i\big) = \frac{1}{n}\sum_{i=1}^{n}\left[\ell(f_{\mathrm{ntk}}(\mathbf{x}_i),y_i) - \ell(y_i,y_i)\right]$$

$$\leq \frac{1}{n}\sum_{i=1}^{n}\left|f_{\mathrm{ntk}}(\mathbf{x}_i) - y_i\right|$$

$$\leq \frac{1}{\sqrt{n}}\sqrt{\sum_{i=1}^{n}\left|f_{\mathrm{ntk}}(\mathbf{x}_i) - y_i\right|^2}$$

$$\leq \frac{\lambda}{c_\sigma^2}\sqrt{\frac{\mathbf{y}^\top(\boldsymbol{\Theta}_\mu^\infty(\mathbf{X},\mathbf{X}) + \lambda/c_\sigma^2\mathbf{I})^{-2}\mathbf{y}}{n}}$$

(2) The next step is to calculate the KL divergence. According to the solution of differential equation in Theorem B.1, we have,

$$\boldsymbol{\theta}_\mu(t) - \boldsymbol{\theta}_\mu(0) = \phi_\mu(\mathbf{x})^\top\big(\boldsymbol{\Theta}_\mu^\infty(\mathbf{X},\mathbf{X}) + \lambda/c_\sigma^2\mathbf{I}\big)^{-1}\big(\mathbf{I} - e^{-(\boldsymbol{\Theta}_\mu^\infty(\mathbf{X},\mathbf{X})+\lambda/c_\sigma^2\mathbf{I})t}\big)\mathbf{y},$$

then $t = \infty$ yields:

$$\boldsymbol{\theta}_\mu(t) - \boldsymbol{\theta}_\mu(0) = \phi_\mu(\mathbf{x})^\top\big(\boldsymbol{\Theta}_\mu^\infty(\mathbf{X},\mathbf{X}) + \lambda/c_\sigma^2\mathbf{I}\big)^{-1}\mathbf{y}$$

Therefore, the KL divergence is,

$$\mathrm{KL} = 1/c_\sigma^2 \cdot \mathbf{y}^\top\big(\boldsymbol{\Theta}_\mu^\infty(\mathbf{X},\mathbf{X}) + \lambda/c_\sigma^2\mathbf{I}\big)^{-1}\boldsymbol{\Theta}_\mu^\infty(\mathbf{X},\mathbf{X})\big(\boldsymbol{\Theta}_\mu^\infty(\mathbf{X},\mathbf{X}) + \lambda/c_\sigma^2\mathbf{I}\big)^{-1}\mathbf{y}$$

$$\leq \frac{1}{c_\sigma^2}\mathbf{y}^\top\big(\boldsymbol{\Theta}_\mu^\infty(\mathbf{X},\mathbf{X}) + \lambda/c_\sigma^2\mathbf{I}\big)^{-1}\mathbf{y}$$

Finally, by Equation 5, we achieve the PAC-Bayesian generalization bound,

$$R_{\mathcal{D}}(Q) \leq \frac{\mathbf{y}^\top\big(\boldsymbol{\Theta}_\mu^\infty(\mathbf{X},\mathbf{X}) + \frac{\lambda}{c_\sigma^2}\mathbf{I}\big)^{-1}\mathbf{y}}{nc_\sigma^2} + \frac{\lambda}{c_\sigma^2}\sqrt{\frac{\mathbf{y}^\top\big(\boldsymbol{\Theta}_\mu^\infty(\mathbf{X},\mathbf{X}) + \frac{\lambda}{c_\sigma^2}\mathbf{I}\big)^{-2}\mathbf{y}}{n}} + O\left(\frac{\log\frac{2\sqrt{n}}{\delta}}{n}\right).$$

$\square$

## C.2    PROOF OF THEOREM 4.5

**Theorem C.2** (Restatement of Theorem 4.5). *Suppose data $S = \{(\mathbf{x}_i,y_i)\}_{i=1}^{n}$ are i.i.d. samples from a non-degenerate distribution $\mathcal{D}$, and $m \geq \mathrm{poly}(n,\lambda_0^{-1},\delta^{-1})$. Consider any loss function $\ell : \mathbb{R} \times \mathbb{R} \to [0,1]$ that is 1-Lipschitz in the first argument such that $\ell(y,y) = 0$. Then with a probability of at least $1 - \delta$ over the random initialization and training samples, the deterministic neural network trained by gradient descent for $T \geq \Omega(\frac{1}{\eta\lambda_0}\log\frac{n}{\delta})$ iterations has population risk $R_{\mathcal{D}}$ that is bounded as follows:*

$$R_{\mathcal{D}} \leq \sqrt{\frac{\mathbf{y}^\top(\boldsymbol{\Theta}_\mu^\infty(\mathbf{X},\mathbf{X}) + \lambda/c_\sigma^2\mathbf{I})^{-1}\mathbf{y}}{n}} + \frac{\lambda}{c_\sigma^2}\sqrt{\frac{\mathbf{y}^\top(\boldsymbol{\Theta}_\mu^\infty(\mathbf{X},\mathbf{X}) + \lambda/c_\sigma^2\mathbf{I})^{-2}\mathbf{y}}{n}} + O\left(\sqrt{\frac{\log\frac{n}{\lambda_0\delta}}{n}}\right).$$

*Proof of Theorem C.2.* In this proof, we use Rademacher-complexity analysis. Let $\mathcal{H}$ be the reproducing kernel Hilbert space (RKHS) corresponding to the kernel $k(\cdot,\cdot)$. It is known that the RKHS norm of a function $f_{\mathrm{ntk}}(\mathbf{x}) = \boldsymbol{\Theta}_\mu^\infty(\mathbf{x},\mathbf{X})\big(\boldsymbol{\Theta}_\mu^\infty(\mathbf{X},\mathbf{X}) + \lambda/c_\sigma^2\mathbf{I}\big)^{-1}\mathbf{y} = \boldsymbol{\alpha}^\top k(\mathbf{x},\mathbf{X})$ is $\|f_{\mathrm{ntk}}\|_{\mathcal{H}} = \sqrt{\boldsymbol{\alpha}^\top k(\mathbf{X},\mathbf{X})\boldsymbol{\alpha}}$, where $k = \boldsymbol{\Theta}_\mu^\infty(\mathbf{X},\mathbf{X})$ and $\boldsymbol{\alpha} = \big(\boldsymbol{\Theta}_\mu^\infty(\mathbf{X},\mathbf{X}) + \lambda/c_\sigma^2\mathbf{I}\big)^{-1}\mathbf{y}$. Then we can bound the $\|f_{\mathrm{ntk}}\|_{\mathcal{H}}$.

$$\|f_{\text{ntk}}\|_{\mathcal{H}} = \sqrt{\mathbf{y}^\top (\boldsymbol{\Theta}_\mu^\infty(\mathbf{X}, \mathbf{X}) + \lambda/c_\sigma^2 \mathbf{I})^{-1} \boldsymbol{\Theta}_\mu^\infty(\mathbf{X}, \mathbf{X})(\boldsymbol{\Theta}_\mu^\infty(\mathbf{X}, \mathbf{X}) + \lambda/c_\sigma^2 \mathbf{I})^{-1} \mathbf{y}}$$

$$\leq \sqrt{\mathbf{y}^\top (\boldsymbol{\Theta}_\mu^\infty(\mathbf{X}, \mathbf{X}) + \lambda/c_\sigma^2 \mathbf{I})^{-1} \mathbf{y}}$$

For function class $\mathcal{F}_B = \{f(\mathbf{x}) = \boldsymbol{\alpha}^\top k(\mathbf{x}, \mathbf{X}) : \|f\|_{\mathcal{H}} \leq B\}$, it is shown that its empirical Rademacher complexity can be bounded as Arora et al. (2019a),

$$\widehat{R}_S(\mathcal{F}_B) = \frac{1}{n} \mathbb{E}\left[ \sup_{f \in \mathcal{F}_B} \sum_{i=1}^n f(\mathbf{x}_i)\gamma_i \right] \leq \frac{B\sqrt{\text{Tr}[k(\mathbf{X}, \mathbf{X})]}}{n}$$

Assume that $\text{Tr}[k(\mathbf{X}, \mathbf{X})] \approx n$. Recall the standard generalization bound from Rademacher complexity, with probability at least $1 - \delta$, we have,

$$\sup_{f \in \mathcal{F}} \left[ \mathbb{E}_{\mathcal{D}}[\ell(f(\mathbf{x}), y)] - \frac{1}{n} \sum_{i=1}^n \ell(f(\mathbf{x}_i), y_i) \right] \leq 2\widehat{R}_S(\mathcal{F}) + 3\sqrt{\frac{\log(2/\delta)}{2n}}$$

There we have,

$$R_{\mathcal{D}} \leq \sqrt{\frac{\mathbf{y}^\top (\boldsymbol{\Theta}_\mu^\infty(\mathbf{X}, \mathbf{X}) + \lambda/c_\sigma^2 \mathbf{I})^{-1} \mathbf{y}}{n}} + \frac{\lambda}{c_\sigma^2} \sqrt{\frac{\mathbf{y}^\top (\boldsymbol{\Theta}_\mu^\infty(\mathbf{X}, \mathbf{X}) + \lambda/c_\sigma^2 \mathbf{I})^{-2} \mathbf{y}}{n}} + O\left( \sqrt{\frac{\log \frac{n}{\lambda_0 \delta}}{n}} \right).$$

$\square$

## D  ADDITIONAL EXPERIMENTS

This section contains additional experimental results. Training is performed with a server with a CPU with 5,120 cores, and a 32 GB Nvidia Quadro V100.

### D.1  VALIDATION OF THEORETICAL RESULTS

We first provide empirical support showing that the training dynamics of wide probabilistic neural networks using the training objective derived from a PAC-Bayes bound are captured by PNTK, which validates Lemma A.6.

Consider a three hidden layer ReLU fully-connected network of the training objective derived from the PAC-Bayesian lambda bound in Equation (5), using an ordinary MSE function as loss. The neural network is trained with a full-batch gradient descent using learning rates equal to one on a fixed subset of MNIST ($|D| = 128$) of ten classifications. A random initialized prior with no connection to data is used since it is in line with our theoretical setting and we only intend to observe the change in parameters rather than the performance of the actual bound.

After $T = 2^{17}$ steps of gradient descent updates from different random initialization, we plot the changes of $\mathbf{W}_\mu^{(l)}$ and $\mathbf{W}_\sigma^{(l)}$ of input/output/hidden layer with respect to width $m$ for each layer on Figure 2. We observe that the relative Frobenius norm change in the input/output layer's weights scales as $1/\sqrt{m}$ while the hidden layers' weight scales is $1/m$ during the training, which verifies Lemma A.6.

### D.2  COMPARISON BETWEEN THEORETICAL BOUNDS AND EMPIRICAL BOUNDS

We make a comparison between theoretical bounds (Equations 14, 15) and empirical bounds. The experiments are performed on two different network structures, a fully connected neural network and a convoluted neural network on MNIST and CIFAR10 datasets. In particular, we build a 3-layer fully-connected neural network with 600 neurons on each layer. The convolutional architecture is equipped with a total of 13 layers with around 10 million learnable parameters. We adopt the same hyper-parameter for both theoretical bounds and empirical bounds. The result is shown in Figure 3. First, for theoretical bounds, we find that the PAC-Bayes bound is smaller than the Rademacher bound. Secondly, we find that both theoretical bounds are larger than empirical bounds, which meets our expectations.

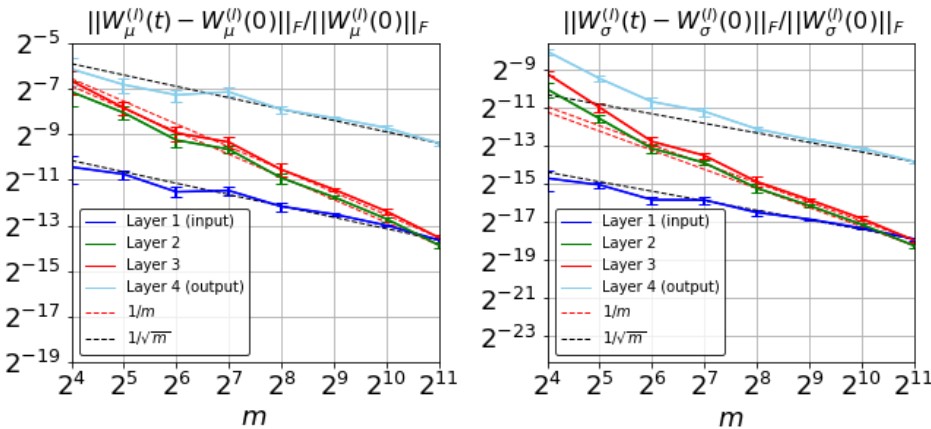

Figure 2: Relative Frobenius norm change in $\mu$ and $\sigma$ respectively during training with MSE loss which is derived from the classic PAC-Bayesian bound, where $m$ is the width of the network.

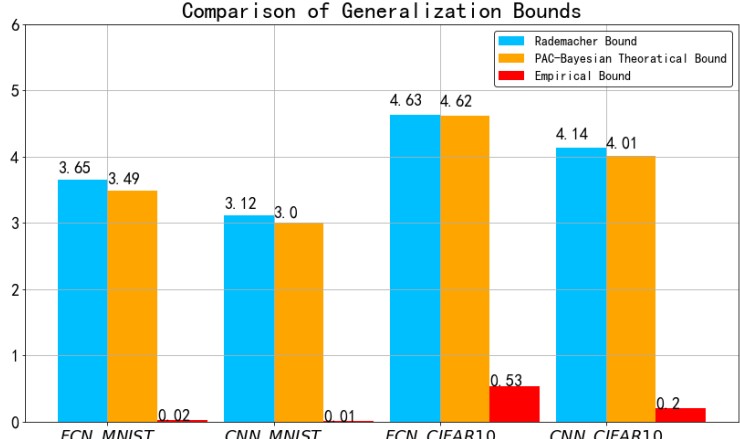

Figure 3: Comparing NTK Rademacher bound, NTK PAC-Bayesian bound and Empirical Bound with different datasets and network structures.

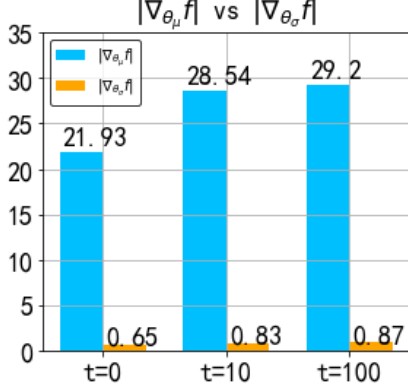

Figure 4: Comparison between the gradient of mean $\mu$ and standard deviations $\sigma$.

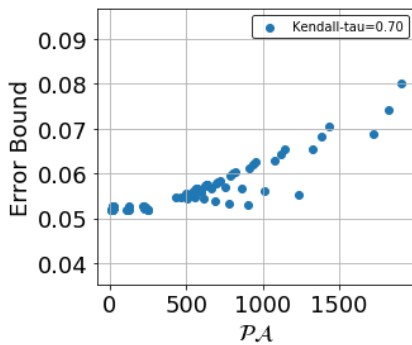

Figure 5: Correlation between aggregated proxy $\mathcal{PA}$ and generalization bound.

### D.3 COMPARISON OF GRADIENT NORM WITH RESPECT TO MEAN WEIGHT AND VARIANCE WEIGHT

We then conduct an experiment to compare the gradient of norm with respect to $\boldsymbol{\theta}_\mu$ and $\boldsymbol{\theta}_\sigma$. The result is shown in Figure 4. We can see that the gradient norm of $\nabla_{\boldsymbol{\theta}_\mu} f(\mathbf{x})$ is much larger than that of $\nabla_{\boldsymbol{\theta}_\sigma} f(\mathbf{x})$, which implies that $\boldsymbol{\theta}_\sigma$ is effectively fixed during gradient descent training.

### D.4 CORRELATION BETWEEN GENERALIZATION BOUND PROXY METRIC AND GENERALIZATION BOUND

In Figure 1, we observe a positive and significant correlation between $\mathcal{PA}$ and generalization bound held among different values of a selected hyperparameter while fixing other hyperparameters. Furthermore, we provide a Figure 5 presenting the correlation for aggregated values of $\rho_0$ and $\lambda$, under the circumstance where 50% data is used for prior training. We can clearly see that lower $\mathcal{PA}$ corresponds to the lower bound, with a strong positive Kendall-tau correlation of 0.7.

### D.5 GRID SEARCH

For selecting hyperparameters, we conduct a grid search over $\rho_0$, percent of prior data, and KL penalty $\lambda$. Notably, we do grid sweep over the data for prior training with different proportion in [0.2, 0.3, 0.4, 0.5, 0.6, 0.7, 0.8, 0.9] since 0.2 is the minimum proportion required for obtaining a reasonably lower value generalization bound (Dziugaite et al., 2020). For the rest, we run over $\rho_0$ at value [0.03, 0.05, 0.07, 0.09, 0.1, 0.3, 0.5, 0.7] for FCN ([0.05, 0.07, 0.09, 0.1, 0.3, 0.5, 0.7, 0.9] for CNN) and KL penalty at [0.0001, 0.0005, 0.001, 0.005, 0.01, 0.05, 0.1, 0.5, 1] for both structures.

