# OpenReview forum: "Demystifying the Optimization and Generalization of Deep PAC-Bayesian Learning"
_ICLR.cc/2023/Conference — Submitted to ICLR 2023_

### Official Review · Reviewer_ag6P · 2022-10-22

**Confidence:** 4
**Correctness:** 3
**Technical Novelty And Significance:** 2
**Empirical Novelty And Significance:** 2
**Recommendation:** 3

**Clarity, Quality, Novelty And Reproducibility:**

**Clarity:**

This paper is well written and easy-to-follow. But the formulation on NTK in page 17 and 18 can be moved to the main text. Besides, the related work has no significant difference on PAC-Bayesian analysis and PAC-Bayesian learning. All of them center around PAC-Bayesian bound.

**Novelty:**

When reading this paper, I thought this paper is quite similar to previous deep learning theory work, e.g., Du et al. (2019), Arova et al. (2019a) in terms of the theoretical results and the technical proofs. Admittedly, the authors mentioned that the probabilistic neural network has another group of parameters, leading to relatively complex derivation. Nevertheless, the contribution in theory and techniques appear not enough and the obtained results incur an extra exponential order of L. In my view, if the authors get rid of the certain reparameterization trick and are able to prove that the weights during the training follow a Gaussian distribution, this will be interesting in both deep learning theory and the probabilistic neural networks community.

Besides, the proposed proxy metric in Eq. (16) based on the label similarity matrix, which has already been discussed in kernel alignment literature. This is able to motivate the authors to have a better discussion.

**Minor issues:**

I only check the derived theoretical results in the appendix and they make sense.
But there are some typos in the derivation, e.g., the last line at the end of page 15; the second inequality in at the top of page 16.


**Strength And Weaknesses:**

**Pros:**

1. Deriving the training dynamics and kernel equivalence for deep and wide probabilistic neural networks under the NTK regime
2. Building a PAC-Bayesian framework for generalization guarantees of probabilistic neural networks

**Cons:**

1. This paper focuses on a certain style of DNNs, where the weights always follow a certain distribution. The authors follow the re-parameterization trick to ensure the weight distribution to be Gaussian during training. I think this is a significant assumption/setting by Eq. (2), which is different from classical deep learning theory literature. In this case, the fourth point in the contribution requires well-polished to avoid some over-claim.

2. In theorem 4.2, the authors forgot a significant assumption: lambda_0(K) > 0. Another drawback is the exponential order the depth L due to the perturbation propagation.

3. More importantly, the matrix in Definition 4.1 is a Jacobian matrix instead of NTK. The authors put the probabilistic NTK into an important position but no formulation of the final PNTK is given. Only \Theta^{\inf} and K^{(l)}_{\inf} in an inner product formulation is not enough until I find it in the appendix at page 17, 18. The paper requires a better organization.

4. Besides, I don’t agree with the authors claiming that PAC-Bayesian bound has an improvement over the Rademacher complexity-based bound. Comparing Eq. (14) and (15), even though the first term in both equations is different, the final convergence rate is the same at the $O(1/\sqrt{n})$ order due to the second term. In fact, Rademacher complexity bound is more general than NTK based bound. This is because, if the neural networks don’t work in a NTK regime, Eq. (14) can not be obtained but Rademacher complexity bound independent of NTK still holds.


**Summary Of The Paper:**

This work theoretically proves the training dynamics of deep and wide probabilistic neural networks and generalization guarantees described by the minimum eigenvalue of the probabilistic NTK. The PAC-Bayesian framework follows the previous classical NTK based, lazy-training results on DNNs, e.g., using the squared loss and gradient descent to prove the kernel equivalence. Besides, a training-free proxy based the derived theoretical results is used for hyper-parameter selection.


**Summary Of The Review:**

In summary, this paper presents the NTK-based analysis for deep and wide probabilistic neural networks in training dynamics and generalization guarantees. Nevertheless, there is not enough enthusiasm to vote for acceptance due to the not significant results and the not new technical proof.

---

> ### Author Response · Authors · 2022-11-18
> **Author Response to Reviewer ag6P, Part II**
>
> 6.  Nevertheless, the contribution in theory and techniques appear not enough and the obtained results incur an extra exponential order of L. In my view, if the authors get rid of the certain reparameterization trick and are able to prove that the weights during the training follow a Gaussian distribution, this will be interesting in both deep learning theory and the probabilistic neural networks community.
>
> We first re-state the technical contribution of this work. Our main target is to characterize the optimization and generalization of  PAC-Bayesian Learning. The core idea of PAC-Bayesian Learning is to adopt PAC-Bayesian bound as an objective function. To understand why PAC-Bayesian Learning can achieve great success, we resort to a theoretical investigation through over-parameterization, in other words, the neural tangent kernel.
>
> Then, we want to kindly point out that the exponential order of the depth L is associated with neural network structure. We may improve exponentially by introducing a residual connection [1] . However, we think this has little to do with the main goal of this work.
>
> Finally, we would like to point out that the reparameterization trick is widely used for training probabilistic neural networks. Including the PAC-Bayes Learning studied in this work that uses re-parametrization to train the PAC-Bayes bound, the optimization of VAE and Bayesian neural networks also use re-parameterization tricks. Given the extensiveness of re-parameterization training methods in optimizing probabilistic neural networks, we claim that our theoretical analysis has a wide range of applications.
>
> 7. Besides, the proposed proxy metric in Eq. (16) based on the label similarity matrix, which has already been discussed in kernel alignment literature. This is able to motivate the authors to have a better discussion.
>
> We agree that our Eq. (16) has a strong connection to the kernel alignment. We thank the reviewer for pointing this out and we have made a discussion about it in the main context:
>
> Note that the proposed proxy metric in Eq. (16) share the same spirit of kernel alignment, a label similarity metric, which has been widely used in the application of deep active learning [2], model selection for fine-tuning [3], and neural architecture search (NAS) [4].
>
> 8. There are some typos in the derivation, e.g., the last line at the end of page 15; the second inequality in at the top of page 16.
>
> Thanks for pointing them out, we have fixed the typos.
>
>
> **References**
>
> [1]  Du, Simon, et al. "Gradient descent finds global minima of deep neural networks." International conference on machine learning. PMLR, 2019.
>
> [2] Haonan Wang, Wei Huang, Andrew Margenot, Hanghang Tong, and Jingrui He. Deep active learning
> by leveraging training dynamics. arXiv preprint arXiv:2110.08611, 2021
>
> [3] Aditya Deshpande, Alessandro Achille, Avinash Ravichandran, Hao Li, Luca Zancato, Charless
> Fowlkes, Rahul Bhotika, Stefano Soatto, and Pietro Perona. A linearized framework and a new
> benchmark for model selection for fine-tuning. arXiv preprint arXiv:2102.00084, 2021.
>
> [4] Jisoo Mok, Byunggook Na, Ji-Hoon Kim, Dongyoon Han, and Sungroh Yoon. Demystifying the
> neural tangent kernel from a practical perspective: Can it be trusted for neural architecture search
> without training? In Proceedings of the IEEE/CVF Conference on Computer Vision and Pattern
> Recognition, pp. 11861–11870, 2022.

---

> ### Author Response · Authors · 2022-11-18
> **Author Response to Reviewer ag6P, Part I**
>
> We sincerely thank the reviewer for recognizing our work is clear and interesting. Below please find the responses to your specific comments and questions.
>
> 1. This paper focuses on a certain style of DNNs, where the weights always follow a certain distribution. The authors follow the re-parameterization trick to ensure the weight distribution to be Gaussian during training. I think this is a significant assumption/setting by Eq. (2), which is different from classical deep learning theory literature. In this case, the fourth point in the contribution requires well-polished to avoid some over-claim.
>
> Thanks for pointing this out. We acknowledge that optimization of probabilistic neural networks is different from that of deterministic neural networks. We have modified the claim of the fourth point in the contribution:
>
> Before: Our technique has a broader applications than prior work (Arora et al., 2019a; Du et al., 2019) because probabilistic neural network has a wider range of application such as ...
>
> After: Our technique of analyzing optimization and generalization of probabilistic neural network through over-parameterization has a wider range of application such as ...
>
> 2. In theorem 4.2, the authors forgot a significant assumption: $\lambda_0(K) > 0$. Another drawback is the exponential order of the depth L due to the perturbation propagation.
>
> Thank you for your comment. We have added the assumption.
>
> We want to kindly point out that the exponential order of the depth $L$ is associated with neural network structure. We may improve exponentially by introducing a residual connection [1]. However, we would like to emphasize that the main aim of this work is to characterize the optimization and generalization of PAC-Bayesian Learning.
>
> 3. More importantly, the matrix in Definition 4.1 is a Jacobian matrix instead of NTK. The authors put the probabilistic NTK into an important position but no formulation of the final PNTK is given. Only \Theta^{\inf} and K^{(l)}_{\inf} in an inner product formulation is not enough until I find it in the appendix at page 17, 18. The paper requires a better organization.
>
> We do think that Definition 4.1 is for PNTK. Expanding the formula in Definition 4.1 with respect to a neural network structure can yield a concrete NTK formula.  We would like to ask the reviewer to point out why you think Definition 4.1 is a Jacobian matrix instead of NTK.
>
> 4. Besides, I don’t agree with the authors claiming that PAC-Bayesian bound has an improvement over the Rademacher complexity-based bound. Comparing Eq. (14) and (15), even though the first term in both equations is different, the final convergence rate is the same at the O(1/\sqrt{n}) order due to the second term. In fact, the Rademacher complexity bound is more general than the NTK based bound. This is because, if the neural networks don’t work in a NTK regime, Eq. (14) can not be obtained but Rademacher complexity bound independent of NTK still holds.
>
> We are very grateful to the reviewer for pointing this out. We acknowledge that the comparison may not rigorously fair and have modified our second contribution to avoid overclaim:
>
> Original claim: Based on the optimization solution, we derive an analytical and guaranteed PAC-Bayesian bound for deep networks for the first time. We find that it is an improvement over the Rademacher complexity bound for a non-probabilistic neural network with a fair comparison.
>
> Modified claim: Based on the optimization solution, we derive an analytical and guaranteed PAC-Bayesian bound for deep networks for the first time. Moreover, our bound differs from other PAC-Bayes bounds. Recent papers require distribution of posterior, while our bound is completely independent of computing the distribution of posterior, as we obtained its analytic result.
>
>
> 5. The related work has no significant difference on PAC-Bayesian analysis and PAC-Bayesian learning. All of them center around PAC-Bayesian bound.
>
> We would like to kindly point out the essential difference between PAC-Bayesian analysis and PAC-Bayesian learning. The PAC-Bayesian analysis uses PAC-Bayes bound to study the generalization property of target models including deep neural networks. On the other hand, PAC-Bayesian learning means training a Bayesian neural network and using a refined PAC-Bayesian bound. The critical difference is that  PAC-Bayesian analysis does NOT use the PAC-bayesian bound as a training objective function.

---

> > ### Comment · Reviewer_ag6P · 2022-11-19
> > **thanks for response**
> >
> > I thank for the response from the authors.
> >
> > Regarding the exponential order of the depth $L$, I'm not sure that the residual connection can avoid this issue. Actually this requires a more detailed discussion to avoid a vacuous analysis on perturbation propagation.
> >
> > Regarding "Definition 4.1 is for PNTK", if one matrix is termed as a NTK matrix, it is able to describe the training dynamics of neural networks by a kernel matrix. That means, the kernel equivalence is needed. Only definition 4.1 is not enough, otherwise an "arbitrary" matrix can be called as a matrix. So in this case, some results in appendix at page 17, 18 are needed to move to the main text.
> >
> > I have read other reviewers' comments on technical novelty, and thus I remain my evaluation.

---

> > > ### Author Response · Authors · 2022-11-20
> > > **Follow up**
> > >
> > > Dear Reviewer ag6P
> > >
> > > We thank the reviewer for your timely response. However, we disagree with your comments.
> > >
> > > (1) Regarding the exponential order of the depth $L$. We point out a possible solution for example residual connection, instead of claiming it can solve the problem. Our opinion is that investigating exponential or polynomial order of perturbation propagation is **out of the scope** of this work, and can be a future work.
> > >
> > > (2) We think Definition 4.1 is the definition of PNTK and **does not** conflict with your description. Could you please point out which results on pages 17, and 18 are needed to move to the main text?
> > >
> > > Best,
> > > Authors of 377

---

### Official Review · Reviewer_Fsnz · 2022-10-25

**Confidence:** 2
**Correctness:** 2
**Technical Novelty And Significance:** 2
**Empirical Novelty And Significance:** Not applicable
**Recommendation:** 3

**Clarity, Quality, Novelty And Reproducibility:**

As mentioned before, the paper lacks clarity. I found it hard to follow the results in this paper and how authors have derived them.

Paper makes some claims that do not make intuitive sense. Thm 4.2 claims for large m empirical risk of probabilistic network converges to zero, i.e. even with perturbation leads to no loss in training error wrt squared loss which seem to suggest the Gaussian should get perfectly concentrated around weights with zero training error. Secondly, when analysing KL divergence term, authors also claim in the theorem B.1 that variance does not change while training. It doesn’t appear to be true and there likely is a mistake in the analysis somewhere.

**Strength And Weaknesses:**

Strengths

- Results obtained are quite involved due to the nature of the convergence analysis

Weaknesses

- Paper lacks clarity; the results obtained are difficult to follow and authors don’t give a palatable proof sketch and defer to the appendix for the detail which is not feasible to review for the conference.

**Summary Of The Paper:**

In this paper authors study wide stochastic networks, where the weights are sampled from a Gaussian with diagonal covariance. Authors aim to give a convergence analysis when the means and variances are obtained via gradient descent trained on the PAC-Bayesian bound objective.

Authors obtain these results by introducing probabilistic neural tangent kernels that explain the dynamics of the training of mean weights and variances.

**Summary Of The Review:**

I recommend rejecting this paper. Main reason for my recommendation is that the results are not clearly written and likely have mistakes in them.

---

> ### Author Response · Authors · 2022-11-18
> **Author Response to Reviewer Fsnz**
>
> We sincerely thank the reviewer for recognizing our work is valuable. Below please find the responses to your specific comments and questions.
>
> 1. Paper lacks clarity; the results obtained are difficult to follow and authors don’t give a palatable proof sketch and defer to the appendix for the detail which is not feasible to review for the conference.
>
> We deeply thank the reviewer for raising an important point regarding clarity. We would like to re-emphasize our proof sketch and defer to the appendix:
>
> * Theorem 4.2 establishes that if the width is large enough, the expected empirical training error converges to zero at a linear rate. To prove Theorem 4.2, we first show that PNTKs at initialization are close to the limiting kernel given the width is large enough. Then we prove the distance between PNTKs and the limiting kernel during training is also bounded, meanwhile loss has a linear convergence rate by induction. Compared to previous work, our network architecture is much more complex (e.g. probabilistic network contains two sets of parameters) and each set involves its own randomness which requires bounding many terms more elaborately. The detailed proof can be found in Appendix A.
>
> * Theorem 4.3 reveals the regularization effect of the KL term in PAC-Bayesian learning, and presents an explicit expression for the convergence result of the output function. The proof of Theorem 4.3 utilizes an argument of linearization of the network model in the infinite width limit. This allows us to obtain an ordinary differential equation for output function with the solution of kernel ridge regression. The details are given in Appendix B.
>
> * Theorem 4.4 establishes a reasonable generalization bound for the PAC-Bayesian learning framework in the large-width condition. We defer the proofs of Theorem 4.4 to Appendix C. Our proof is based on a characterization of the empirical error and KL divergence term via the explicit solution found in Theorem 4.3.
>
> The details can be found in Section 5 of the main context. Please do let us know if anything is still unclear to you. We are happy to give further explanations during the discussion period.
>
> 2. Paper makes some claims that do not make intuitive sense. Thm 4.2 claims for large m empirical risk of probabilistic network converges to zero, i.e. even with perturbation leads to no loss in training error wrt squared loss which seem to suggest the Gaussian should get perfectly concentrated around weights with zero training error. Secondly, when analysing KL divergence term, authors also claim in the theorem B.1 that variance does not change while training. It doesn’t appear to be true and there likely is a mistake in the analysis somewhere.
>
> Thanks for your questions. For the first question regarding Thm 4.2, we understand your concern that when the loss gets zero the variance of weights should be zero. We would like to point out that the loss is defined as follows:
>
> where we take the expectation over the posterior $Q$. And in this work, the $Q$ is the randomness in the variance weight, according to the reparametrization. This implies that even if the variance is not zero, we can still obtain a zero train error.
>
> The claim in the theorem B.1 also uses the definition of loss $\hat R_S(Q) = E_{f∼Q} \hat R_S(f)$ and interchanged integration and differentiation, which can be found in the proof of Proof of Lemma A.6. on Page 23.

---

### Official Review · Reviewer_7Cjm · 2022-10-27

**Confidence:** 3
**Correctness:** 3
**Technical Novelty And Significance:** 2
**Empirical Novelty And Significance:** 2
**Recommendation:** 5

**Clarity, Quality, Novelty And Reproducibility:**

The problem setup is very clear. I had difficulty following the technical results. The main issue is that results do not come with intuition and the connection between the theorems is not clear.

**Strength And Weaknesses:**

The problem setup in the paper is very interesting. However, I found the paper quite difficult to follow. The connection between theorems is not very clear. More importantly, the author did not discuss how their proof technique is differ from the NTK results.

Questions:

1- I think equation 3 only works if the loss function is 0-1. Later in the paper the squared loss is used. Could you clarify this point?

2- What does $+- \xi$ in Eq. (13) mean?

3- What is the intuition behind Equation (16)?

4- In the numerical results section, why is there no plot on the actual generalization versus the theoretical bounds?

5- In Thm 4.2 the KL term is dropped and the GD is only on the empirical risk. Later, in Thm 4.3 the authors also consider the KL term. I don't quite understand the intuition behind the proof technique here.

**Summary Of The Paper:**

This paper considers the problem of learning a probabilistic neural network using the regularizer that comes from the PAC-Bayes generalization bounds. The main assumption is that the weights are sampled from Gaussian distribution with given mean and variance. The training algorithm is based on applying the gradient descent on the mean and variance of each weight. The training loss function is based on the PAC-Bayes bound which consists of an empirical risk and KL term. Then, in the limit that the width goes to infinity the authors provide a closed-form expression on the output of the neural network similar to NTK results.  The author also provides some numerical results which show the effectiveness of their approach.

**Summary Of The Review:**

The paper considers an important problem: using PAC-Bayes objective to train a probabilistic neural network. The contribution of the results is not clear in terms of the differences between NTK literature.

---

> ### Author Response · Authors · 2022-11-18
> **Author Response to Reviewer 7Cjm**
>
> We sincerely thank the reviewer for recognizing our work is valuable and interesting. Below please find the responses to your specific comments and questions.
>
> 1. I think equation 3 only works if the loss function is 0-1. Later in the paper the squared loss is used. Could you clarify this point?
>
> We use squared loss to train the probabilistic neural network (PNN) but use a general or a suitable loss $\ell \in [0, 1]$ to evaluate PNN’s generalization. The training loss and evaluation loss can be different. For example, we use 0-1 loss to evaluate PNN’s generalization but usually do not use it to train PNN. In particular, we use the generalization bound (Eq. 6) as the objective function to train. After we get the solution of the output function, we plug it back into Eq (5) and use the loss function $\ell$ to obtain a generalization bound.
>
> 2. What does  $\pm \mathcal{E}$ in Eq. (13) mean?
>
> Thanks for your question. Basically, $\mathcal{E}$ measures the difference between output function of a finite-width probabilistic neural network and an infinitely-wide probabilistic neural network whose result can be formulated by a kernel regression. The exact definition of $\mathcal{E}$ can be found in appendix B.
>
> 3. What is the intuition behind Equation (16)?
>
> We derive Equation (16) from our theoretical generalization bound Eq (14). The intuition is that Eq (14) differs from other PAC-Bayes bounds. Recent papers require distribution of posterior, while our bound is completely independent of computing the distribution of posterior, as we obtained its analytic result. This implies that Eq (14) can be measured in a training-free manner which can save a lot of time compared with other training-dependent methods. Furthermore, the generalization bound has a strong correlation with a final performance. Inspired by these reasons, we develop a training-free proxy to efficiently select hyper-parameters.
>
> 4. In the numerical results section, why is there no plot on the actual generalization versus the theoretical bounds?
>
> We are deeply grateful for the reviewer's constructive suggestion and we totally agree that it is important to show the difference between actual generalization and the theoretical bounds numerically. We conduct experiments on MNIST and CIFAR-10. The results are shown in Figure 3 of Appendix D.2. We have the following findings:
>
> * Firstly, the PAC-Bayes bound is smaller than the Rademacher-based bound.
>
> * Secondly, we find that both theoretical bounds are larger than empirical bounds.
>
> In summary, it is not surprising to see that theoretical bounds are larger than actual generalization bounds.
>
> 5. In Thm 4.2 the KL term is dropped and the GD is only on the empirical risk. Later, in Thm 4.3 the authors also consider the KL term. I don't quite understand the intuition behind the proof technique here.
>
> The proof sketch is in a divide-and-conquer manner. If we consider the full objective function (including the KL divergence term), we have two challenges: 1)  the probabilistic networks contain two sets of parameters, and each set involves its own randomness which requires bounding many terms, making the analysis for empirical risk difficult. 2) To compute the KL divergence term, we need to obtain the explicit result of the posterior distribution.
>
> We thus adopt a divide-and-conquer manner to solve this problem. We first study the empirical risk and find that the solution is close to a kernel regression with PNTK being its kernel. Then we extend the NTK-based result to the objective function with KL divergence.

---

### Official Review · Reviewer_DZ2U · 2022-10-27

**Confidence:** 3
**Correctness:** 3
**Technical Novelty And Significance:** 3
**Empirical Novelty And Significance:** 2
**Recommendation:** 5

**Clarity, Quality, Novelty And Reproducibility:**


Please see the ''Strength And Weaknesses'' section.

**Strength And Weaknesses:**


## Strength
1. Overall, this paper is well-organized and written clearly.
2. The proofs of the theoretical results seem correct although I have not checked them line-by-line.
3. The obtained theory results are valuable if it is the first time for deep probabilistic neural networks.
4. Based on the theory results, the proposed proxy measure for efficient hyperparameter selection is effective with experimental support.



## Weaknesses
1. As highlighted for deterministic models, the analysis framework based on the neural tangent kernel has limitations, such as it cannot characterize the feature learning process in deep learning. This paper also has these limitations, which are for probabilistic models.
This paper does not discuss these limitations and more discussions should be added.
2. The technical novelty may be limited since the NTK-based analyses have been widely used in deep learning theory although this paper considers the probabilistic neural networks.
3. In my view, the comparisons between obtained the PAC-Bayesian generalization bound and the Rademacher complexity-based one might not be fair because of the following reasons. First, the definitions of the generalization errors for these two are different. Second, from these two bounds, although we
can find the key differences are the first term, as the authors claimed, one is for $O(\frac{1}{n})$, and another is $0(\frac{1}{\sqrt{n}})$, but these bounds totally depending on $o(\frac{1}{\sqrt{n}})$. Further, the authors claimed that the obtained PAC-Bayesian bound is non-vacuous in the experiments.
Thus, I recommend the authors do additional experiments to compare these two bounds and related discussions should be added.


**Summary Of The Paper:**


This paper aims to theoretically investigate the optimization and generalization of deep PAC-Bayesian learning in the setting of wide neural networks. Technically, based on
the analysis ideas of the neural tangent kernel (NTK) for the deterministic models under the overparameterized settings, this paper characterizes the convergence of the probabilistic neural
networks, which builds the connection with the kernel ridge regression via the probabilistic neural tangent kernel (PNTK). Then, the authors give the PAC-Bayesian generalization bound,
which is an improvement over the Rademacher complexity-based bound. Further, inspired by the obtained bound, they propose a proxy measure for efficient hyperparameter selection. Finally, some experimental results are also
provided.

**Summary Of The Review:**

In summary, this paper theoretically characterizes the optimization and generalization of deep probabilistic neural networks via the analytical framework of the neural tangent kernel. It is valuable if it is the first time to provide these results.
However, this paper needs some necessary revisions as in the 'Weaknesses' part.

---

> ### Author Response · Authors · 2022-11-18
> **Author Response to Reviewer DZ2U, Part II**
>
> 4. Further, the authors claimed that the obtained PAC-Bayesian bound is non-vacuous in the experiments. Thus, I recommend the authors do additional experiments to compare these two bounds and related discussions should be added.
>
> We are deeply grateful for the reviewer's constructive suggestion and we totally agree with the reviewer that it is important to show the difference between the two bounds numerically. Following the suggestion, we conduct experiments on MNIST and CIFAR10 showing the PAC-Bayes bound is better than Rademacher-based bound. The results are shown in Figure 3 of Appendix D.2. We have the following findings:
>
> * Firstly, the PAC-Bayes bound is smaller than the Rademacher-based bound.
>
> * Secondly, we find that both theoretical bounds are larger than empirical bounds.
>
> In summary, we believe that PAC-Bayes bound can behave better than Rademacher-based bound since it enjoys faster convergence for the KL term.
>
> **References**
>
> [1] Damian, Alexandru, Jason Lee, and Mahdi Soltanolkotabi. "Neural networks can learn representations with gradient descent." Conference on Learning Theory. PMLR, 2022.
>
> [2] Ba, Jimmy, et al. "High-dimensional Asymptotics of Feature Learning: How One Gradient Step Improves the Representation." arXiv preprint arXiv:2205.01445 (2022).
>
> [3] Arora, Sanjeev, et al. "Fine-grained analysis of optimization and generalization for overparameterized two-layer neural networks." International Conference on Machine Learning. PMLR, 2019.
>
> [4] Du, Simon, et al. "Gradient descent finds global minima of deep neural networks." International conference on machine learning. PMLR, 2019.

---

> ### Author Response · Authors · 2022-11-18
> **Author Response to Reviewer DZ2U, Part I**
>
> We sincerely thank the reviewer for recognizing our work is valuable, clear, and inspiring. Below please find the responses to your specific comments and questions.
>
> 1. As highlighted for deterministic models, the analysis framework based on the neural tangent kernel has limitations, such as it cannot characterize the feature learning process in deep learning. This paper also has these limitations, which are for probabilistic models. This paper does not discuss these limitations and more discussions should be added.
>
> Thanks for your comments. We acknowledge the limitations of neural tangent kernel methods, as they can only characterize the lazy training dynamics of deep neural networks. In other words, NTK-based techniques cannot capture the feature learning process for both deterministic models [1,2] and probabilistic neural networks. However, we would like to kindly point out that the goal of this paper is not to try to capture the feature learning for probabilistic neural networks, but we aim at providing **new important convergence and generalization analysis** for PAC-Bayesian learning. Nevertheless, we thank you for your comment and we add a discussion to clarify this point in the Discussion section.
>
> 2. The technical novelty may be limited since the NTK-based analyses have been widely used in deep learning theory although this paper considers the probabilistic neural networks.
>
> Thank you for the comment. We would like to make a few clarifications on our contributions and the technical novelty below:
>
> * Firstly, our convergence analysis is on a much more complex network architecture (e.g. probabilistic networks contain two sets of parameters) than prior work [3,4], which is also more challenging than previous work because it requires bounding many terms in this work differently and more elaborately due to each set involving its own randomness.
>
> * Secondly, we highlight that our theoretical results hold for **multi-layer** probabilistic neural networks, thus our results have broad applications such as the Variational Auto-encoder and deep Bayesian networks. Besides, we would like to point out the fact that the final solution of PAC-Bayesian Learning in the infinite-width limit is kernel ridge regression is also quite novel, since this result is obtained for PAC-Bayes Learning for the first time. We believe our technique can provide the basis for the analysis of over-parameterized probabilistic neural networks.
>
> To sum up, we have developed new convergence and generalization analysis which deepens our understanding of PAC-Bayesian Learning . Our future work is to push the analysis to feature learning using probabilistic neural networks.
>
> 3. In my view, the comparisons between obtaining the PAC-Bayesian generalization bound and the Rademacher complexity-based one might not be fair because of the following reasons. First, the definitions of the generalization errors for these two are different. Second, from these two bounds, although we can find the key differences are the first term, as the authors claimed, one is for O(1/n), and another is O(1/sqrt{n}) but these bounds totally depending on O(1/sqrt{n}).
>
> We are very grateful to the reviewer for pointing this out. We acknowledge that the overall dependence is on O(1/sqrt{n}), thus we have modified our second contribution to avoid overclaim:
>
> Original claim: Based on the optimization solution, we derive an analytical and guaranteed PAC-Bayesian bound for deep networks for the first time. We find that it is an improvement over the Rademacher complexity bound for a non-probabilistic neural network with a fair comparison.
>
> Modified claim: Based on the optimization solution, we derive an analytical and guaranteed PAC-Bayesian bound for deep networks for the first time. Moreover, our bound differs from other PAC-Bayes bounds. Recent papers require distribution of posterior, while our bound is completely independent of computing the distribution of posterior, as we obtained its analytic result.
>
> To show the **improvement** of the PAC-Bayesian bound over the Rademacher complexity bound, we conducted experiments to measure the theoretical bound on MNIST and CIFAR-10. We provide a detailed description in the next answer.

---

### Author Response · Authors · 2022-11-28
**General Response**

We thank all the reviewers for their helpful feedback on the submission! Based on the valuable comments and suggestions, we have better presented the contribution of our work, and the meanings of our theoretical analysis. And we provided new results regarding comparision between theoretical bounds and empricial bounds (Figures 3 in Appendix D.2) to the latest draft (text highlighted in blue).

We’ve provided replies to individual reviewer comments. Please let us know if you have additional questions or need further clarifications. Thanks again!

---

### Decision · Program_Chairs · 2023-01-20

**Decision:**

Reject

**Justification For Why Not Higher Score:**

Two lines of work may be pursued to improve the paper:
- clarify how the results can move towards more actionable deep models;
- better explain and discuss the theoretical entailments of the results:
-- dependence on the overparametrization
-- exponentional L-dependence
-- comparison with other bounds.

**Justification For Why Not Lower Score:**

Still a promising piece of contribution, an attempt to provide theoretical results to the realm of deep learning.

**Metareview: Summary, Strengths And Weaknesses:**

This paper provides a PAC-Bayesian learning approach to a deep and wide probabilistic neural networks, an approach that makes the connection between the training objectives derived from PAC-Bayes bounds and the Neural Tangen Kernel (NTK) learning dynamics in over-parameterized setting.

Strengths of the paper, as noted by the reviewers:
+ theoretical analysis on the learning dynamics of neural networks
+ PAC-Bayesian bound-based learning algorithm
- but the analysis works for Probabilistic (stochastic?) Neural Networks and not deterministic ones
- the statements of the results may be hard to grasp at times
- there are imprecisions on how to read the results (cf. Rademacher-centered comparison with the bound)